# Polarization conversion in bottom-up grown quasi-1D fibrous red phosphorus flakes

Zhaojian Sun[1,3], Wujia Chen[1,3], Bowen Zhang[1], Lei Gao[1], Kezheng Tao[1], Qiang Li[1], Jia-Lin Sun [2] & Qingfeng Yan [1] ✉

Fibrous red phosphorus (RP) has triggered growing attention as an emerging quasi-one-dimensional (quasi-1D) van der Waals crystal recently. Unfortunately, it is difficult to achieve substrate growth of high-quality fibrous RP flakes due to their inherent quasi-1D structure, which impedes their fundamental property exploration and device integration. Herein, we demonstrate a bottom-up approach for the growth of fibrous RP flakes with (001)-preferred orientation via a chemical vapor transport (CVT) reaction in the $P/Sn/I_2$ system. The formation of fibrous RP flakes can be attributed to the synergistic effect of Sn-mediated $P_4$ partial pressure and the $SnI_2$ capping layer-directed growth. Moreover, we investigate the optical anisotropy of the as-grown flakes, demonstrating their potential application as micro phase retarders in polarization conversion. Our developed bottom-up approach lays the foundation for studying the anisotropy and device integration of fibrous red phosphorus, opening up possibilities for the two-dimensional growth of quasi-1D van der Waals materials.

Being a representative allotrope of elemental phosphorus, red phosphorus (RP) has garnered sustained interest due to its merits of structural diversity, easy availability and low cost since its initial discovery in 1840s[1]. RP exists in five distinct modifications, namely Form-I (amorphous RP), Form-II, Form-III, Form-IV (fibrous RP), and Form-V (violet RP)[2]. Among these, fibrous RP is a newly recognized quasi-one-dimensional (quasi-1D) crystal composed of parallel-arranged phosphorus atomic chains weakly bounded in bundles by van der Waals forces[3]. Recent experimental studies[4] revealed its promising potential in electronic devices with high mobility reaching about 300 cm² V⁻¹ s⁻¹. More significantly, fibrous RP shows giant linear/nonlinear optical anisotropy, which may find potential device applications in phase-matching elements, polarizers, sensing, and polarization-sensitive photodetectors[5]. Fibrous RP with a unique quasi-1D structure exhibits splendid electronic and optical properties, making it a hopeful candidate for next-generation optoelectronic devices. However, the in-depth study of this new quasi-1D van der Waals crystal and its device application hinge on access to the high-quality fibrous RP flakes.

Unfortunately, due to its inherent quasi-1D structure, fibrous RP tends to exist in the form of bundles or rods, as observed in the previous reports[4,6–13], and it is difficult to obtain fibrous RP flakes. Moreover, it is of paramount importance to directly grow fibrous RP flakes on diverse target substrates to meet the various characterization and device application requirements. Although the top-down exfoliation technique can be employed to prepare fibrous RP on a substrate from its bulk single crystals, it normally appears in non-flake forms[5,12–15], suffering from poor reproducibility, uncontrollable morphology and potential contamination, which hinders the fundamental properties exploration and micro device integration. Thus, direct bottom-up growth of high-quality fibrous RP flakes on diverse substrates is highly desired.

Moreover, extensive research has been conducted in recent years to investigate the optical anisotropy of two-dimensional (2D) van der Waals (vdW) crystals[16–20]. High-symmetry 2D vdW materials, such as $MoS_2$[16] and h-BN[17], exhibit significant interlayer optical anisotropy in the infrared and visible regions, respectively, while they are optically isotropic within the plane. However, when it

[1]Engineering Research Center of Advanced Rare Earth Materials (Ministry of Education), Department of Chemistry, Tsinghua University, 100084 Beijing, P. R. China. [2]Department of Physics, Tsinghua University, 100084 Beijing, P. R. China. [3]These authors contributed equally: Zhaojian Sun, Wujia Chen. ✉e-mail: yanqf@mail.tsinghua.edu.cn

comes to low-dimensional materials with nanoscale thickness, harnessing the interlayer anisotropy becomes challenging. In-plane anisotropic materials with low symmetry, such as ReS₂[18], ReSe₂[18], and black phosphorus (BP)[18–21], offer only moderate in-plane birefringence. Consequently, by reducing the dimensionality from 2D to quasi-1D, one can expect to achieve significant optical anisotropy along the interchain and intrachain directions. However, only a few quasi-1D materials, including Te[22], Sb₂Se₃[23], and BaTiS₃[24], have demonstrated strong in-plane optical anisotropy through experimental investigations. Therefore, it is crucial to explore the in-plane optical anisotropy of high-quality fibrous RP flakes, as it holds great importance and potential for various applications.

Herein, we achieved the growth of high-crystallinity fibrous RP flakes on diverse substrates with (001)-preferred orientation via a chemical vapor transport (CVT) reaction. The growth mechanism of fibrous RP flakes was investigated in depth through experiments and theoretical calculations, which can be attributed to the Sn-mediated P₄ partial pressure and the SnI₂ capping layer-directed growth. A comprehensive investigation was conducted into the in-plane optical anisotropy of fibrous RP flakes using angle-resolved polarized Raman spectroscopy (ARPRS), polarization-resolved visible light spectroscopy, and polarization-resolved optical microscopy (PROM). Additionally, we employed polarization-resolved monochromatic transmitted light intensity mapping to visually display the potential application of fibrous RP flakes directly grown on transparent quartz substrates as micro phase retarders for polarization conversion.

## Results

### Substrate growth of high-quality fibrous RP flakes

Fibrous RP flakes with high crystalline quality were directly grown on a SiO₂/Si substrate by a CVT reaction with an inverse temperature gradient, as schematically illustrated in Fig. 1a. In a typical experiment, commercial amorphous red phosphorus, Sn and I₂, placed in the source zone (low-temperature end), were partially transported to the growth zone after running the established temperature program (Supplementary Figs. 1a, b) and subsequently, fibrous RP flakes were grown on the SiO₂/Si substrate at the growth zone (high-temperature end) during the cooling process. Simultaneously, the residual precursor contributed to the formation of bulk BP crystals at the source zone (Supplementary Figs. 1c and 2). Fibrous RP flakes possess a unique quasi-1D structure, which is composed of parallel-arranged atomic chains weakly bounded in bundles by van der Waals forces along the *a*-axis and *c*-axis (Fig. 1b). While along the *b*-axis, the building block[P8]P2[P9]P2is covalently bonded with each other (Supplementary Fig. 3). Figure 1c shows a typical optical image of fibrous RP flakes on the SiO₂/Si substrate, exhibiting the regular rectangular shape with a lateral size of about 6–8 µm. The thickness of the corresponding flake is 37 nm as measured by atomic force microscope (AFM, Fig. 1d). Supplementary Fig. 4a displays more AFM images of as-grown fibrous RP flakes, and their thickness mainly ranges from 20 to 200 nm (Supplementary Fig. 4b). The nonuniformity in thickness is likely attributed to unsteady gaseous precursors supply[25]. Figure 1e depicts the distinctive Raman peaks of fibrous RP, which is consistent with previous reports[13]. Complicated Raman vibration modes are attributed

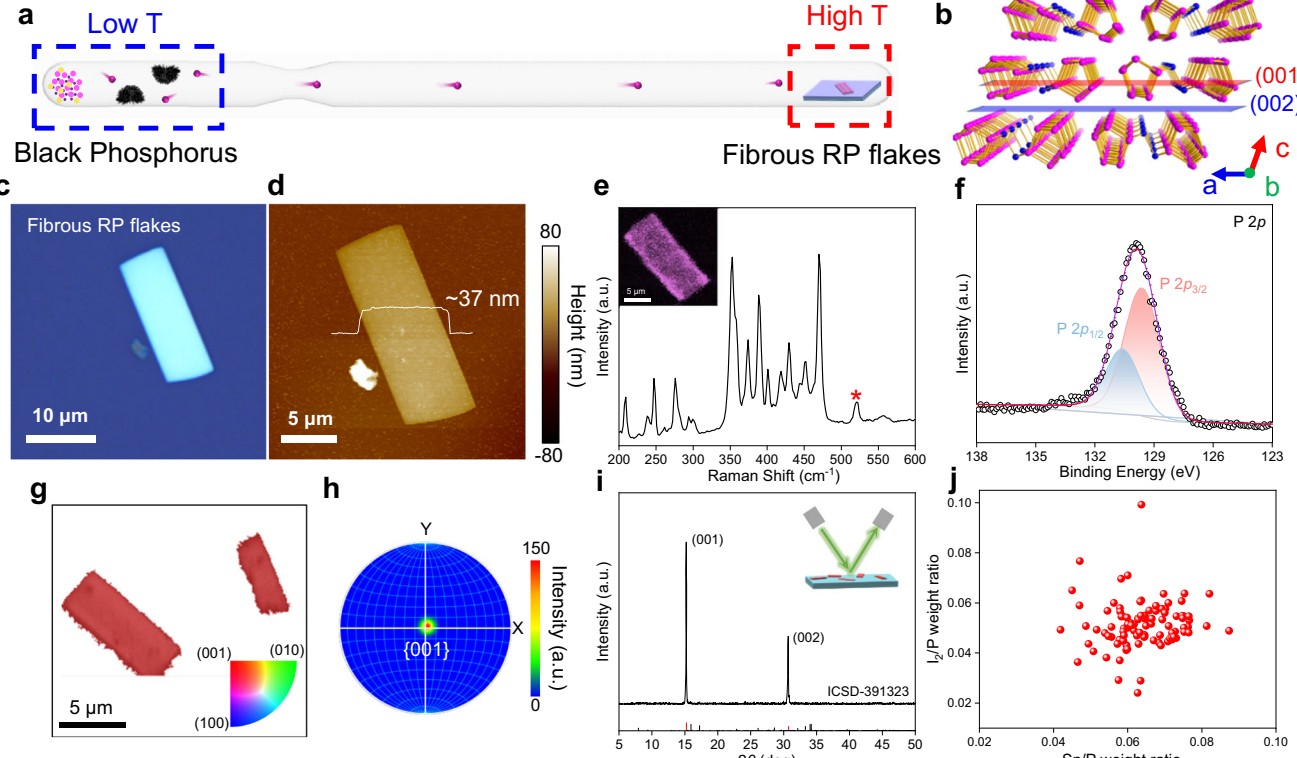

**Fig. 1 | Growth and characterization of fibrous red phosphorus (RP) flakes. a** Schematic illustration for chemical vapor transport (CVT) growth of fibrous RP flakes on a SiO₂/Si substrate via an inverse temperature gradient in the P/Sn/I₂ system. **b** Crystal structure of fibrous RP viewed from the [010] direction and the (001)/(002) diffraction planes for fibrous RP denoted by red and blue planes, respectively. **c** Typical optical image of fibrous RP flakes directly grown on the SiO₂/Si substrate. **d** Atomic force microscope (AFM) image of the corresponding fibrous RP flake, showing a thickness of ~37 nm. **e** Raman spectrum of fibrous RP flakes measured under 532 nm excitation. The peak labeled "*" originates from the SiO₂/Si substrate. The unit of intensity is arbitrary units (a.u.). Inset: Raman mapping at ~471 cm⁻¹ of the fibrous RP flake. **f** High-resolution X-ray photoelectron spectroscopy (XPS) spectrum of P 2*p* for fibrous RP flakes directly grown on the SiO₂/Si substrate. The red and blue fitting curves correspond to P 2*p*₃/₂ and P 2*p*₁/₂, respectively. **g** Inverse pole figure (IPF) of fibrous RP flakes along the *z*-axis. Inset: The color diagram for IPF. **h** Pole figure of fibrous RP flakes for the {001} plane group. **i** X-ray diffraction (XRD) pattern of fibrous RP flakes directly grown on the SiO₂/Si substrate. The inset displays the measurement configuration. **j** The statistical growth conditions of 100 successful attempts to obtain fibrous RP flakes.

to its low symmetry and the variety of atoms in the unit cell. Raman mapping and photoluminescence (PL) mapping for the fibrous RP flake exhibit uniform signals across the whole flake, demonstrating high continuity and homogeneity of the as-prepared fibrous RP flake (Fig. 1e, inset, and Supplementary Fig. 5). Furthermore, X-ray photoelectron spectroscopy (XPS) was performed to analyze the valence of P element (Fig. 1f and Supplementary Fig. 6). The fitted doublet at 129.6 and 130.6 eV corresponds to P $2p_{3/2}$ and P $2p_{1/2}$, respectively, exactly confirming the existence of elemental P without other valence states.

To further identify the preferred orientation of fibrous RP flakes on the $SiO_2$/Si substrate, X-ray diffraction (XRD) and electron backscattered diffraction (EBSD) were carried out. Figure 1i presents two sharp diffraction peaks merely, indexed as (001) and (002) planes (Fig. 1b) which verifies the high crystallinity and preferred growth orientation of the fibrous RP flakes. Meanwhile, the homogeneous colors for inverse pole figure along the z-axis (IPF-Z, Fig. 1g) visually confirm its preferred (001) growth orientation as well as high crystallinity uniformity, which is in agreement with XRD results (Fig. 1i). And the pole figure of fibrous RP flakes for {001} plane group displays a single intense spot at the center of stereographic projection (Fig. 1h). Specifically, during the growth of fibrous RP flakes, perturbation of gas phase species supply leads to the formation of steps in the vertical direction, which may appear inclined at certain angles. Nevertheless, the top exposed surface consistently maintained a preferred orientation of (001), as depicted in Supplementary Fig. 7. Subsequently, density functional theory (DFT) was adopted to calculate the surface energy of (100), (010), and (001) planes for fibrous RP, as presented in Supplementary Table 1. The (001) plane demonstrates the lowest surface energy (8.27 meV·$Å^{-2}$), delivering a stable atomic configuration, which accounts for the (001)-preferred orientation of fibrous RP flakes. The growth conditions of fibrous RP flakes on the $SiO_2$/Si substrate were further investigated, and it was found that the weight ratios of precursors play a dominating role. As depicted in Fig. 1j, each of the red balls corresponds to a pair of P/Sn and P/$I_2$ weight ratios leading to the successful growth of fibrous RP flakes, where the weight ratios are calculated based on 400 mg of precursor amorphous RP. The dense distribution of the red balls implies a narrow growth window for fibrous RP flakes. Moreover, we found that fibrous RP flakes could crystallize across the step edges on the back side of a $SiO_2$/Si substrate and quartz ampoule wall readily (Supplementary Fig. 8). This phenomenon inspired us that the growth process is likely to be attached to the substrate simply rather than epitaxial growth. As expected, after extensive attempts, fibrous RP flakes were successfully grown on quartz (0001), Si (111), GaN (0001), sapphire (0001), mica, and soda-lime glass as well (Supplementary Figs. 9 and 10). The substrate-independent growth behavior of high-quality fibrous RP flake benefits the exploration of its abundant intrinsic properties that were inaccessible before.

The crystal structure and crystallinity of fibrous RP flakes were further determined by transmission electron microscopy (TEM). A fibrous RP flake sample in plan-view was acquired by a PMMA (polymethyl methacrylate)-assisted transfer method[26] (see "Methods" for details). The typical low-magnification TEM image displays a near rectangle shape with two long parallel sides (Fig. 2a). Corresponding lattice fringe with an interplanar spacing of 0.32 nm is in good accordance with the (04$\bar{1}$) plane of fibrous RP (Fig. 2b). Furthermore, the indexing results in fast Fourier transform image (Fig. 2c) are well in line with the simulated electron diffraction pattern along the zone axis [$\bar{4}\,\bar{5}\,2\bar{0}$] (Supplementary Fig. 11). Therefore, according to the crystallographic orientation, fibrous RP flake is b-axis-aligned along the long parallel sides, as indicated by the pink dashed line in Fig. 2a, which is attributed to the high bonding tendency of (010) plane with the highest surface energy (33.60 meV·$Å^{-2}$, Supplementary Table 1).

Atomic-resolution cross-sectional High Angle Angular Dark Field-Scanning Transmission Electron Microscopy (HAADF-STEM) was

performed to confirm the b-axis-aligned orientation profoundly. In particular, focused ion beam milling (FIB, see "Methods" for details) was used to prepare a cross-sectional TEM sample in a given direction which is perpendicular to the long parallel sides of the rectangular flake, as shown in Supplementary Fig. 12. Figure 2d presents a low-magnification cross-sectional TEM image and the bright sharp diffraction spots suggest exceedingly high crystallinity of the fibrous RP flake (Fig. 2e). The low-magnification cross-sectional HAADF-STEM image and the corresponding elemental mappings (Fig. 2f–i) further indicate the distinct fibrous RP/$SiO_2$ interface. Moreover, the high-resolution cross-sectional HAADF-STEM image in Fig. 2j demonstrates a perfectly layered structure, and the (001) plane with the layer-to-layer spacing of 0.58 nm is parallel to the $SiO_2$/Si substrate, which is well consistent with the aforementioned XRD and EBSD characterizations. Meanwhile, the pentagonal atomic arrangement of parallel twin tubes in Fig. 2j matches well with the structural model viewed from the b-axis (Fig. 1b), corroborating its b-axis-aligned orientation along the long parallel sides thoroughly.

## CVT growth mechanism of fibrous RP flakes

It is known that fibrous RP can be prepared in the P/$I_2$ system via the CVT reaction, which tends to appear in bulk crystals form or nanostructured form (urchin-like bundles or nanowires) due to the uncontrollable $P_4$ molecules supply and anisotropic growth behavior[4,8,9,13]. Therefore, the key to synthesizing fibrous RP flakes lies in two aspects. The first is to achieve a controllable $P_4$ molecules supply. The second is to manipulate the growth behavior of fibrous RP. In this work, by introducing Sn into the P/$I_2$ system, forming the P/Sn/$I_2$ system, both of the two goals can be attained, enabling the formation of fibrous RP flakes.

From a thermodynamic standpoint, the driving force of the phase transition is the chemical potential difference between the metastable phase and the stable phase[27]:

$$\Delta\mu = RT\ln\left(\frac{p}{p_0}\right), \qquad (1)$$

where R is the gas constant, T is the absolute temperature, p is the supersaturated vapor pressure and $p_0$ is the saturated vapor pressure. After nucleation, the growth rate of each crystal facet is closely related to the relationship between the chemical potential difference $\Delta\mu$ and kinetic barrier ($\Delta G$), which determines the diversity of crystal growth morphology[28–31]. Generally, $\Delta G$ is inversely proportional to the surface energy ($\gamma$)[30–32]. According to the surface energy of fibrous RP crystal calculated by DFT (Supplementary Table 1), $\gamma_{(010)} \gg \gamma_{(100)} > \gamma_{(001)}$. Consequently, $\Delta G_{(010)} \ll \Delta G_{(100)} < \Delta G_{(001)}$. Therefore, the kinetic barrier is lowest along the tube (b-axis) for fibrous RP, while the kinetic barriers along the a-axis and c-axis are higher and closer together (Fig. 3a). The relative magnitude between the driving force and the growth kinetic barrier can be theoretically altered by adjusting the synthesis temperature (T) or controlling the $P_4$ partial pressure (p), which enables the formation of fibrous RP with desired morphology. In a prior report[33], it was shown that Te, which has a similar quasi-1D structure, can achieve morphology control of 1D rods, 2D flakes, and 3D spheres by adjusting the synthesis temperature (T). However, interconversion occurs between Form II, Form IV (fibrous RP), and Form V (violet phosphorus) of various crystalline red phosphorus at similar temperatures[34,35]. As a result, controlling the morphology of fibrous RP by adjusting the synthesis temperature merely is not feasible. It has been proved that BP crystals can be grown in a P/Sn/$I_2$ system through CVT reaction[36]. Therefore, we introduced Sn into the P/$I_2$ system, which was used to synthesize bulk or nanostructured fibrous RP crystals previously[4,8,9,13]. In this case, BP tends to gain an advantage in competition with fibrous RP during phase transition in the CVT reaction, consuming most of the available phosphorus source[36]. As a

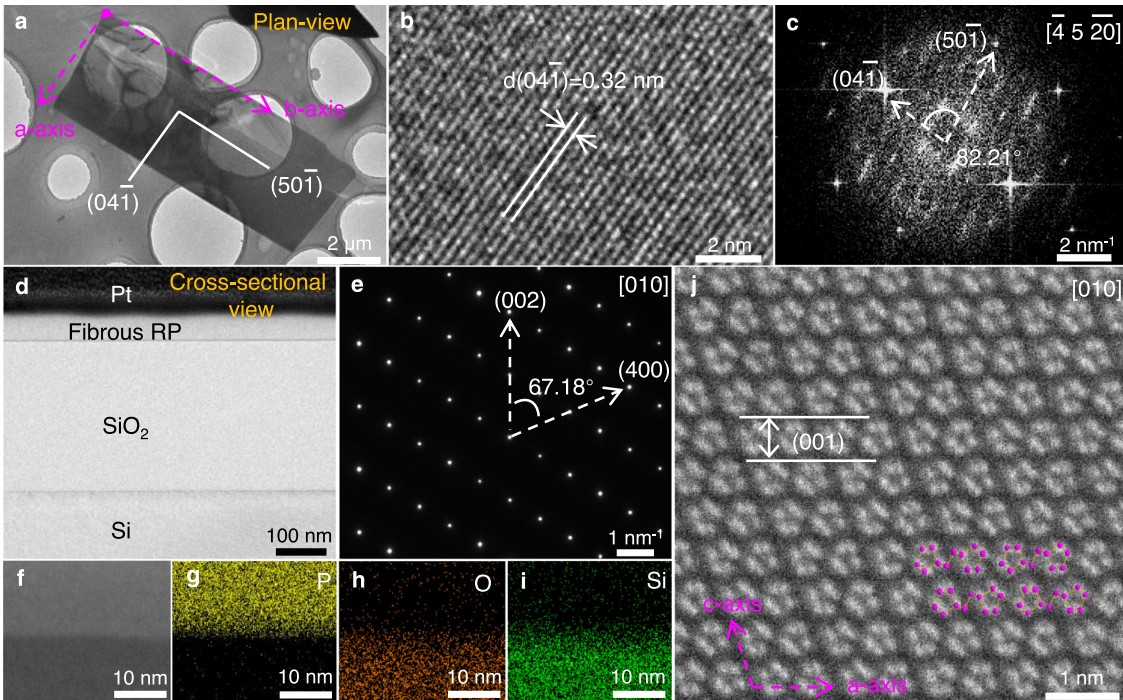

**Fig. 2 | Crystal structure characterization of fibrous RP flakes. a** Typical plan-view Transmission electron microscopy (TEM) image of the fibrous RP flake acquired by poly-methyl methacrylate (PMMA) assisted transfer processes. The pink dashed line and the white solid line indicate the crystal orientation. **b** Plan-view high-resolution transmission electron microscopy (HRTEM) image of the fibrous RP flake. *d* denotes the interplanar spacing of the corresponding lattice fringe. **c** The electron diffraction pattern obtained from fast Fourier transform based on (**b**). **d** Low-magnification cross-sectional TEM image and **e** corresponding selected area electron diffraction (SAED) pattern of the fibrous RP flake. **f** Low-magnification cross-sectional high angle angular dark field-scanning transmission electron microscopy (HAADF-STEM) image of the fibrous RP flake on the $SiO_2$/Si substrate and the corresponding elemental mappings of **g** P, **h** O, and **i** Si. **j** Atomic-resolution cross-sectional HAADF-STEM image of the fibrous RP flake, showing a distinct layered structure of (001) plane with 0.58 nm spacing. The pink circles indicate the structural model of fibrous RP viewed from the *b*-axis.

result, the amount of Sn added can be used to mediate the $P_4$ partial pressure in the $P/Sn/I_2$ system and balance the growth of BP crystals at the source zone and fibrous RP crystals at the growth zone. Additionally, an inverse temperature gradient was adopted to hinder the transfer of $P_4$ molecules to the substrate, leading to a further reduction in the partial pressures of $P_4$ molecules at the growth zone.

As a result, when the $P_4$ partial pressure maintains very low, $P_4$ molecules tend to grow along the longitudinal direction, resulting in the formation of fibrous RP nanowires or nanorods (Fig. 3b, i); with a slight increase in $P_4$ partial pressure, lateral growth becomes favorable, leading to the formation of fibrous RP flakes (Fig. 3b, ii); when $P_4$ partial pressure further increases, isotropic rapid growth occurs, and fibrous RP polycrystalline spheres form (Fig. 3b, iii). It is worth mentioning that under low $P_4$ partial pressure in the $P/Sn/I_2$ system, fibrous RP 1D rods (few) and 2D flakes (most) are observed simultaneously (Supplementary Fig. 13a, b). However, in the $P/I_2$ system, without competitive consumption of most $P_4$ molecules through the growth of BP crystals, only bulk fibrous RP polycrystalline spheres form owing to the high $P_4$ partial pressure in the growth zone (Supplementary Fig. 13c). Therefore, one important role of Sn in the $P/Sn/I_2$ system is to balance the competitive consumption of $P_4$ molecules in the CVT reaction, facilitating the formation of fibrous RP flakes at a low $P_4$ partial pressure at the high-temperature end.

Apart from Sn-mediated $P_4$ partial pressure regulating the morphology of fibrous RP, $SnI_2$ also plays a crucial role in the formation of fibrous RP flakes. Gas phase species in the $P/Sn/I_2$ system were calculated using CalPhaD methods, which have proven to be a powerful tool for investigating gas phase transport reactions[37]. Figure 3c indicates that the partial pressure of $P_4$ and $SnI_2$ at equilibrium is at least two orders of magnitude higher than the competitive molecules (i.e., $SnI_4$ and $I_2$), suggesting that $SnI_2$ may contribute to the growth of fibrous RP

flakes. To further investigate the role of $SnI_2$ in the formation mechanism of fibrous RP flakes, we employed Gaussian to calculate the stable configuration of fibrous RP intermediates, using the energy of starting materials as the reference. Due to the unique bonding configuration of fibrous RP, there is likely an addition–elimination equilibrium of $SnI_2$ molecules during its growth along the tube. Similar to the formation of black phosphorus[37], Sn atoms in $SnI_2$ may coordinate with phosphorus atoms at the edge of the parallel twin tubes. The coordination of double P9 subunit with four $SnI_2$ entities to *i* (Fig. 3d) is electronically favored by a total energy difference of $\Delta E_{tot} = -658$ kJ/mol. Then, in order to lower the energy of the system, this bond between phosphorus and tin will be broken and reformed, allowing the addition of $P_4$ molecules along the tube to yield *ii* (Fig. 3d). We also calculated the intermediate of fibrous RP without $SnI_2$ capping (Fig. 3d, iii). In this premise, fibrous RP is expected to grow spontaneously with successive addition of $P_4$ atoms, leading to the growing preference along the *b*-axis. Yet it is not the case. The energy of this configuration is slightly higher by 319 kJ/mol compared to the $SnI_2$-capped structure, indicating that the capping of $SnI_2$ is energetically advantageous and more promising in $P/Sn/I_2$ system (Fig. 3e). Since the $SnI_2$ capping layer significantly stabilizes fibrous RP and inhibits its growth along the *b*-axis, it further promotes the growth of parallel phosphorus tubes along the direction of van der Waals forces, favoring the formation of 2D flakes with low aspect ratio rather than 1D rods/wires with high aspect ratio.

To check whether the $SnI_2$ capping layer still exists or not after the formation of fibrous RP flakes, we conducted a detailed analysis of the element distribution along the depth of the fibrous RP surface using time-of-flight secondary ion mass spectrometry (ToF-SIMS). Our results indicate that there were no detectable $I^+$ signals and a weak $Sn^+$ signal in the selected ion bombardment area. This is

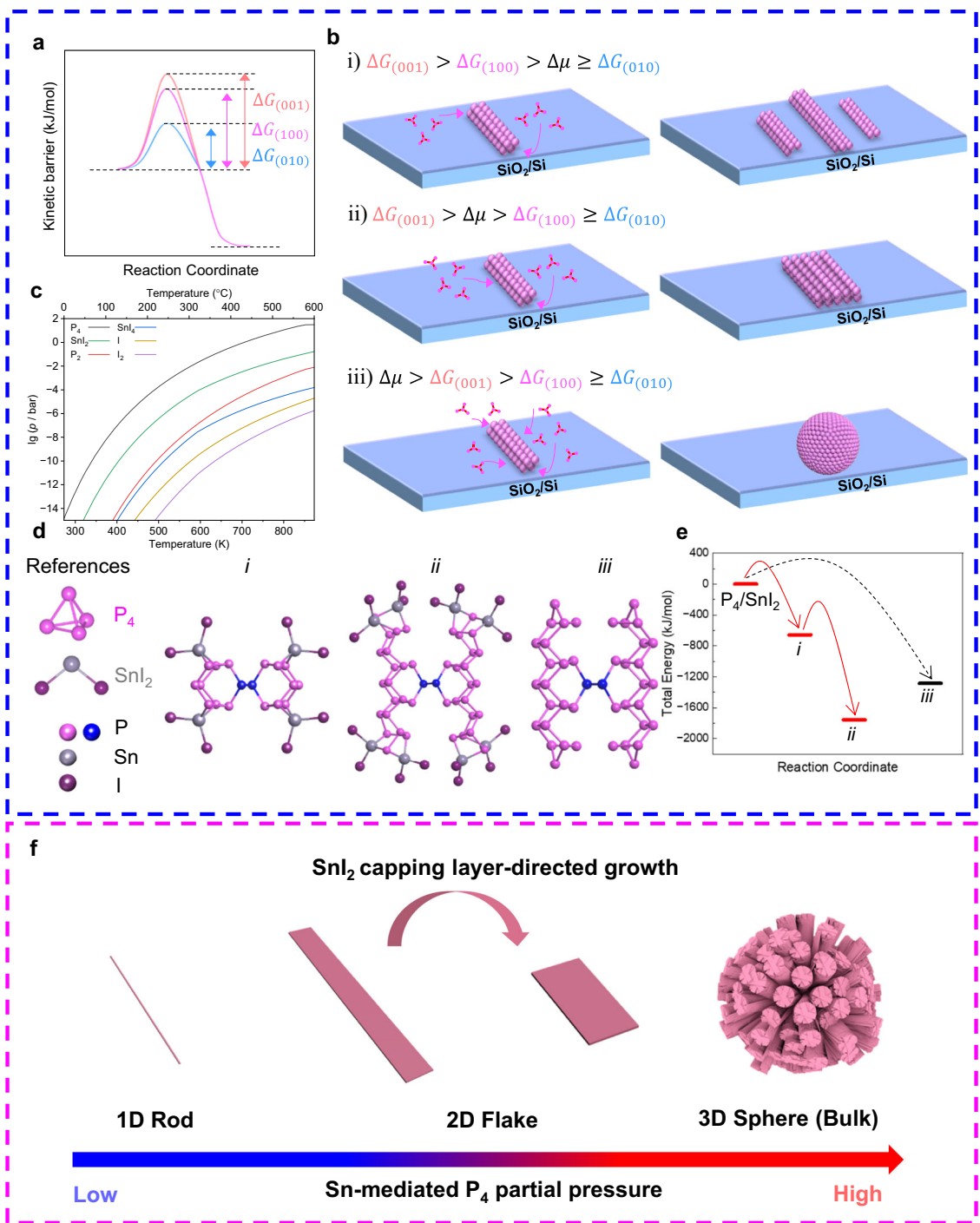

**Fig. 3 | Growth mechanism of fibrous RP flakes. a** Schematic diagram of the kinetic barrier $\Delta G$ of (001), (010), (100) crystal facets of fibrous RP. **b** Schematic illustration for the morphology evolution of fibrous RP according to the relationship between chemical potential difference and kinetic barrier. The morphology evolution can be divided into three types, (i) $P_4$ molecules tend to grow along the longitudinal direction; (ii) $P_4$ molecules tend to grow along the longitudinal and lateral directions; (iii) $P_4$ molecules tend to grow in all directions. The pink arrows indicate the direction of $P_4$ molecule binding. **c** Partial pressure diagram of the corresponding equilibrium gas pressure of the $P/Sn/I_2$ system in the temperature range from 273 to 873 K calculated by CalPhaD. **d** Schematic diagram of the starting materials ($P_4$, $SnI_2$) and geometry-optimized structures (*i, ii, iii*) in the proposed growth mechanism. **e** Total energy of the starting materials ($P_4$, $SnI_2$) and geometry-optimized structures (*i, ii, iii*) calculated by Gaussian. Note: Data used in (**e**) are summarized in Supplementary Tables 2 and 3. **f** Schematic diagram of the growth mechanism fibrous RP crystals.

because, during the cooling process, the capping $SnI_2$ undergoes decomposition into Sn and $I_2$ again. While the former remains on the substrate surface, the latter diffuses randomly and can exist at any position within the ampoule. Specifically, we observed five peaks that correspond to five isotopes of tin, namely $^{116}Sn$, $^{117}Sn$, $^{118}Sn$, $^{119}Sn$, and $^{120}Sn$ (Supplementary Fig. 14a, b). It is worth noting

that fibrous RP often exists as high mass-to-charge ratio phosphorus cluster fragments after being bombarded[38], which are beyond the detection range of the device. Consequently, the signal of low mass-to-charge ratio phosphorus ion is weak (Supplementary Fig. 14c–f). Furthermore, the depth distribution image revealed that Sn was only present on the substrate surface, and there was no signal

distribution for detecting Sn or I inside the sample (Supplementary Figs. 14c, g, h).

We also checked the feasibility of $I_2$ as a capping layer. As indicated by the CalPhaD calculations, in the absence of Sn, the main gas phase species with the most significant partial pressure are $P_4$ and $I_2$ (Supplementary Fig. 15a). Although the structure capped with iodine also converges, it is bonded to the edge in the form of a single iodine atom (Supplementary Fig. 15b), which is clearly not the main species in the gas phase. Moreover, the structure capped with iodine deviates from the original configuration of fibrous RP. Thus, we can conclude that in this scenario, iodine serves solely as a traditional CVT transport agent and does not have any capping effect. Experimentally, we created a low partial pressure region of $P_4$ in the $P/I_2$ system under the same temperature conditions by reducing the amount of red phosphorus and controlling mass transfer. As anticipated, we were able to obtain stacked, high aspect ratio fibrous RP flakes that were hundreds of microns long under low $P_4$ pressure without the presence of Sn (Supplementary Fig. 16). However, it should be noted that the crystal quality, orientation, and thickness of these flakes are not comparable to the fibrous RP flakes with low aspect ratio that are capped with $SnI_2$. Overall, the synergistic effect of Sn-mediated $P_4$ partial pressure and the $SnI_2$ capping layer-directed growth facilitates the formation of fibrous RP flakes (Fig. 3f).

### In-plane optical anisotropy of fibrous RP flakes

Theoretically, threefold rotational symmetry breaking endows fibrous RP with giant anisotropy between intrachain and interchain[5,24]. Furthermore, unlike uniaxial crystals such as graphene[39] and $MoS_2$[40], which exhibit in-plane optical isotropy with their optical axis perpendicular to the plane, fibrous RP in the low-symmetry triclinic system possesses a triaxial ellipsoid optical indicatrix, resulting in significant in-plane optical anisotropy. As a result, we conducted a comprehensive investigation into the in-plane optical anisotropy of as-grown fibrous RP flakes.

To reveal the in-plane phonon vibrational anisotropy of fibrous RP, angle-resolved polarized Raman spectroscopy (ARPRS) was performed on fibrous RP flakes grown on the $SiO_2/Si$ substrate under both parallel-polarization and cross-polarization configurations (Fig. 4a). In such measurements, the analyzer direction is parallel or perpendicular to the incident light polarization direction. We defined the angle between the incident light polarization direction and the direction perpendicular to the $b$-axis of fibrous RP as the polarized angle $\theta$ (Supplementary Fig. 17a). Figure 4b, c exhibits the contour-color map of polarized Raman intensity for fibrous RP flake under parallel- and cross-polarization configuration, respectively. For these two configurations, the Raman peak intensity of almost every vibrational mode varies significantly with a period of 180°. The main typical Raman peaks (209, 248, 353, 375, 390, and 471 cm$^{-1}$) of fibrous RP flake exhibit a 2-lobed shape with the maximum intensity at 0° and 180° for parallel-polarization configuration, while at 90° and 270° for cross-polarization configuration (Supplementary Figs. 18 and 19). However, the Raman vibration peak at 390 cm$^{-1}$ exhibits a strong signal and significant change in the parallel configuration, whereas the signal is weak and shows little change in the cross-configuration (Fig. 4b, c and Supplementary Figs. 19, 22, 25, 28, 30 and 32). This discrepancy can be attributed to the difference in the Raman tensor element of fibrous RP in the two configurations at 390 cm$^{-1}$. Notably, we observed that the direction along the maximum Raman intensity is perpendicular to the $b$-axis for parallel polarization while parallel to the $b$-axis for cross-configuration. Thickness-dependent ARPRS was measured at various excitation wavelengths as well (Supplementary Figs. 20–32). The obtained polar plots for the typical Raman peak of 471 cm$^{-1}$ are summarized in Supplementary Table 4, which exhibits orientation-dependent Raman polarization as well. When we altered the azimuth of the sample, it was still observed (Fig. 4d, e). It is worth mentioning

that this phenomenon has been found in highly oriented pyrolytic graphite (HOPG) and carbon nanotube, which is described as the antenna effect[41,42]. Meanwhile, angle-resolved polarized PL spectroscopy was performed under the same configurations. In both configurations, the PL peak intensity reaches a maximum at 90° and minimum at 0 and 180°, which exhibits the orientation-dependent PL intensity as well (Supplementary Fig. 33). Above results demonstrate that the strong in-plane anisotropy of phonon vibration and photoluminescence exists in fibrous RP, which can be contributed by its singular quasi-1D structure.

Moreover, polarization-resolved visible light spectroscopy was performed to investigate the in-plane optical anisotropy. To ensure the $b$-axis of the fibrous RP flake was parallel to the $x$-axis, we placed it horizontally and defined the angle between the direction of incident light and the $b$-axis as $\varphi$. As $\varphi$ varies from 0 to 90°, the optical contrast of the fibrous RP flake showed significant changes, as seen in the optical images (Fig. 4f, I, II). The optical contrast of the fibrous RP flake on a $SiO_2/Si$ substrate can be defined as[43]:

$$C = \frac{I_{FRP} - I_{substrate}}{I_{substrate}}, \qquad (2)$$

where $I_{FRP}$ and $I_{substrate}$ represent the reflectance intensities of the fibrous RP flake and $SiO_2/Si$ substrate, respectively. Figure 4g, h depicts the polarization-resolved optical contrast spectra for I and II, respectively. The dominant peak in the optical contrast spectra (~560 nm) results in a green appearance of the fibrous RP flakes when the incident light direction is parallel to the $b$-axis. Conversely, when the incident light direction is perpendicular to the $b$-axis, the peak at ~600 nm of the optical contrast spectra dominates, giving the flakes a red appearance.

In addition to the anisotropic optical contrast, Fig. 4i, j and Supplementary Fig. 34 illustrate the polarization-resolved reflection, transmission and absorption spectra of the horizontally placed fibrous RP flake. The reflection spectra were measured with reference to a silver mirror. All the optical spectra of the fibrous RP exhibited substantial differences when the light direction was parallel to the $b$-axis (0°) and perpendicular to it (90°), demonstrating the strong in-plane optical anisotropy of the flake. It is worth noting that the sum of transmittance and reflectance of the same sample may exceed 1 due to the disparity in the configuration of the reflected and transmitted light paths, as well as the variance in reference light intensity. As a result, the absorption spectrum was independently measured instead of being extracted by combining the reflection and transmission spectra.

The aforementioned polarization-resolved visible light spectra originated from the in-plane optical refractive index difference caused by the low symmetry of fibrous RP. PROM was utilized to further investigate the anisotropic refraction of fibrous RP flakes. Figure 4k shows the transmitted images of fibrous RP flakes under crossed polarized light. Specifically, the analyzer was set perpendicular to the incident linearly polarized light, and the transmitted images of the fibrous RP flakes at different rotation angles were captured by a charge-coupled device (CCD) camera. Normally, the CCD cannot detect the transmitted light, resulting in a dark image. However, as the incident linearly polarized light passes through the fibrous RP flake, the polarization state undergoes a transformation into elliptically polarized light due to the birefringence effect. This transformation is not completely suppressed after passing through the analyzer, resulting in a bright image. With the rotation of fibrous RP flakes, the image displays periodic changes. The brightness of the image depends on the position of the sample relative to the incident polarized light. When the incident light aligns with the fast or slow axis of the fibrous RP flakes, its polarization state remains unchanged, resulting in the darkest optical image. Conversely, when the incident light is at a 45° angle from the slow or fast axis, its polarization state undergoes the

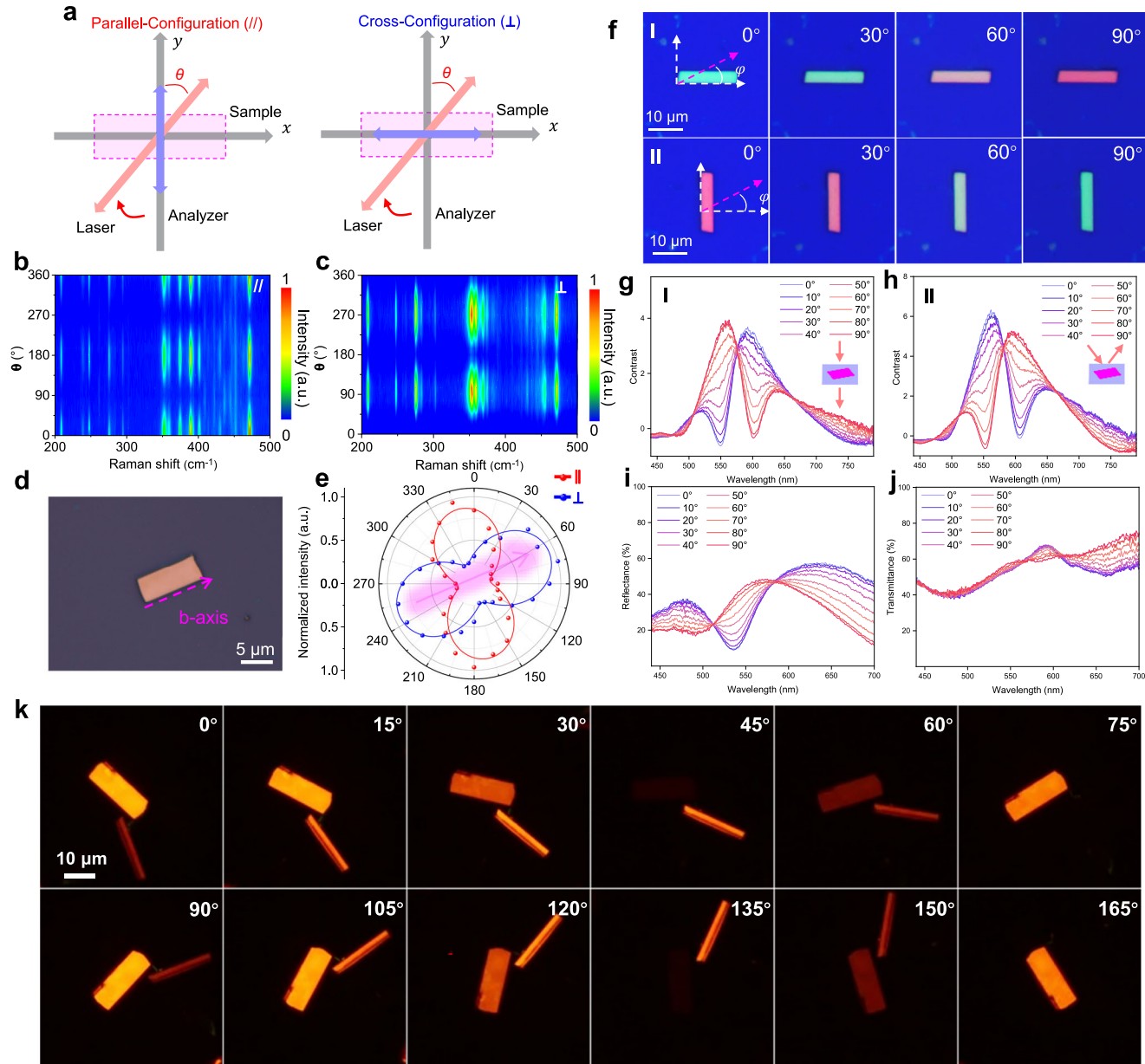

**Fig. 4 | In-plane optical anisotropy of the substrate-grown fibrous RP flakes.**
**a** Schematic diagram of parallel-polarization and cross-polarization configurations.
We defined the angle between the incident light polarization direction and the
direction perpendicular to the $b$-axis of fibrous RP as $\theta$. Contour plot of polarized
Raman intensity under **b** parallel (//) and **c** cross (⊥) polarization configuration.
**d** Optical image of the fibrous RP flake. **e** Polar plots for the typical Raman mode
(471 cm⁻¹) under parallel (red curve) and cross (blue curve) polarization config-
uration corresponding to the fibrous RP flakes in (**d**). Dots are experimental data,
and the solid lines are the fitting curves. The pink arrow indicates the direction of
the $b$-axis of the fibrous RP flakes in (**d**). **f** Optical images of fibrous RP irradiated by
polarized white light. We defined the angle between the incident light polarization
direction and the $b$-axis of fibrous RP as $\varphi$. Polarization-resolved optical contrast
spectra of **g** horizontally (I) and **h** vertically placed (II) fibrous RP flakes. Keeping
the sample position unchanged, $\varphi$ varies from 0 to 90° during the measurement.
**i**, **j** Polarization-resolved reflection spectra and transmission spectra of the fibrous
RP flakes, respectively. The test condition is consistent with (**g**), (**h**). **k** Transmitted
images of fibrous RP flakes under crossed polarized light. The scale bar applies to
all the images in (**k**).

most change, resulting in the brightest optical image[44]. Thus, the
PROM measurement can clearly display the in-plane anisotropic
refraction of fibrous RP flakes.

**Fibrous RP flake micro phase retarder**
Based on the in-plane anisotropic refraction of fibrous RP flakes
observed above, the dispersion relation of refractive index $n$ along
different axes of fibrous RP flake were calculated by DFT. Since fibrous
RP crystallizes in the triclinic crystal system ($P\bar{1}$ space group), its
crystallographic coordinate system is oblique to the rectangular
coordinate system and the optical principal axis coordinate system

(Supplementary Fig. 35). In the rectangular coordinate system where
the $x$-axis is parallel to the $a$-axis of fibrous RP flake, the refractive index
calculated by DFT can be expressed as a second-order symmetric
tensor with cross terms:

$$A = \begin{bmatrix} n_{xx} & n_{xy} & n_{xz} \\ n_{yx} & n_{yy} & n_{yz} \\ n_{zx} & n_{zy} & n_{zz} \end{bmatrix} \quad (n_{ij} = n_{ji}). \quad (3)$$

Through coordinate transformation, the symmetric tensor can be
diagonalized to obtain its expression in the principal axis coordinate

system:

$$\Lambda = \begin{bmatrix} n_1 & 0 & 0 \\ 0 & n_2 & 0 \\ 0 & 0 & n_3 \end{bmatrix}, \tag{4}$$

where $n_1$, $n_2$ and $n_3$ represent the principal refractive index along the optical principal axis ($n_1 < n_2 < n_3$). It is worth noting that fibrous RP possesses considerable anisotropy of refractive index $n$ along the optical principal axis, which behaves as a positive biaxial crystal (Supplementary Fig. 36a). In the high absorption region, two peaks of birefringence ($\Delta n = n_3 - n_1$) are observed, with values of 1.646 at 416 nm and 1.382 at 592 nm, respectively. Subsequently, there is a sharp decline followed by a consistent trend in the birefringence values, which remains at approximately 0.5 in the near-infrared region. This behavior can be attributed to the intricate interaction between light and solids in the high absorption region, where processes such as photon absorption and electron transitions contribute to a greater degree of anisotropy in the refractive index. Furthermore, the xOy-plane refractive index differences ($\Delta n' = n_y - n_x$) exhibit attractive characteristics, with a value of 0.739 at 467 nm and consistently large values (above 0.3) across a wide spectral range (see Supplementary Fig. 36b).

Giant birefringence, especially large in-plane refractive index difference and substrate-independent growth behavior of fibrous RP flakes, render them a promising material choice to construct micro phase retarders for polarization state conversion. In a polarization conversion process, the incident polarized light vector is decomposed into two linearly polarized light vectors with perpendicular vibration directions along the slow axis and fast axis of a crystal with birefringence (Supplementary Fig. 37); as schematically illustrated in Fig. 5a, the optical paths of the two beams of light propagating in the fibrous RP crystal are no longer equal due to its in-plane refractive index difference, thus generating phase retardation and further achieving polarization state conversion. Specifically, the light vector of the transmitted light **E** and phase retarder value $\delta$ can be expressed by the following equation[21,45]:

$$\mathbf{E} = E_0 \cdot \cos\theta \cdot t_x \cdot \cos(\omega t) \cdot \mathbf{x} + E_0 \cdot \sin\theta \cdot t_y \cdot \cos(\omega t - \delta) \cdot \mathbf{y}, \tag{5}$$

$$\delta \equiv \frac{(n_y - n_x) 2\pi d}{\lambda}, \tag{6}$$

where $E_0$ is the amplitude of incident light, $\theta$ is the angle between the incident light direction and the fast axis, $t_x$, $t_y$ are the transmission coefficients, $\omega$ is the frequency, **x**, **y** are the unit vectors along the fast and slow axis, $d$ is the thickness of the crystal, $\lambda$ is the wavelength of the incident light, $n_x$, $n_y$ are the xOy-plane refractive index, respectively.

Polarization-resolved monochromatic transmitted light intensity mapping was carried out to visually demonstrate the polarization state conversion in a fibrous RP flake. Briefly, we utilized 532 nm linearly polarized light (horizontal direction) to irradiate the fibrous RP flake grown on a transparent quartz substrate in the transmission mode and collected the mapping of transmitted light intensity within the test region at different analyzer angles. The transmitted light intensity mappings of the transparent quartz substrate are depicted in Supplementary Fig. 38, which reveals that the intensity reaches the maximum intensity at 0° and complete extinction at 90°. It demonstrates the homogeneity of the substrate, indicating that it does not alter the polarization state of the incident polarized light. As illustrated in Fig. 5b, the fibrous RP with a thickness of 64 nm within the blue box exhibits consistent changes in the transmitted light intensity along with the substrate. This indicates that the intensity undergoes complete extinction at 90°, similar to the substrate behavior. However, for

the fibrous RP with a thickness of 596 nm within the pink box, the transmitted light intensity maintains a certain value even when the substrate undergoes complete extinction at a 90° (Fig. 5c and Supplementary Fig. 39). As shown in Fig. 5d, f, the polarization-resolved intensity of input/output spectra was well fitted with Supplementary Equation 7. We define the degree of polarization (DOP) as $(I_{max} - I_{min})/(I_{max} + I_{min})$, where $I_{max}$, $I_{min}$ are the maximum light intensity and minimum light intensity, respectively. For the thin sample (64 nm), the output spectrum closely matches the input spectrum, consistent with the mapping results. Additionally, the DOP value is 0.995, indicating a minimal change in the polarization state of the incident light (Fig. 5e). Conversely, for the thick sample (596 nm) with a DOP of 0.8, the incident linearly polarized light is transformed into elliptically polarized light (Fig. 5g). The difference in the DOP values is primarily influenced by the phase retarder value $\delta$ and the angle $\theta$ between the incident light direction and the fast axis, as explained in Supplementary Equation 12. Theoretically, as the crystal thickness increases, the corresponding phase retarder value also increases, leading to a smaller DOP value. Specifically, when the phase retarder value is held constant, the DOP demonstrates a correlation with $\theta$. At $\theta = 0°$ or 90°, the DOP reaches its maximum value of 1, indicating an unchanged polarization state. To demonstrate the polarization state conversion of incident light using fibrous RP flakes of varying thicknesses, we observed that the DOP of the transmitted light is 0.981, 0.916, and 0.873 for fibrous RP flakes with thicknesses of 100 nm, 147 nm, and 178 nm, respectively (Supplementary Figs. 41–43). Additionally, we determined the corresponding phase retarder value $\delta$ and the in-plane refractive index difference $\Delta n'$. The summarized results can be found in Supplementary Table 5. Consequently, through phase retardation estimation, we calculated $\Delta n'$ to be $0.26171 \pm 0.02359$. It's important to note that this value is only half of the theoretical value. Several factors contribute to this discrepancy. First, theoretical calculations tend to overestimate the refractive index. Second, uncertainties arising from factors like the hypothesis incorporated in the formula derivation and systematic errors in signal acquisition during the testing process can impact the accuracy of $\Delta n'$ obtained through phase retardation estimation. These factors highlight the complexities involved in obtaining precise refractive index values. In conclusion, the polarization state of transmitted light can be controlled by the thickness of fibrous RP flakes, making it useful for various photonic processes.

## Discussion

In summary, we achieved the direct bottom-up growth of fibrous RP flakes on a rich variety of inert substrates, including SiO$_2$/Si, quartz, Si, GaN, sapphire, mica, as well as soda-lime glass via a CVT reaction in the P/Sn/I$_2$ system. The obtained highly crystalline fibrous RP flakes with (001) preferred plane growth orientation were verified by XRD, EBSD and TEM. Furthermore, atomic-resolution cross-sectional HAADF-STEM was implemented to determine that the crystallographic $b$-axis is parallel to the long sides of fibrous RP flakes. Through both theoretical calculation and experimentation, the growth mechanism of fibrous RP flakes has been unveiled, and it can be attributed to two key factors. First, the competitive consumption of the phosphorus source through Sn-mediated CVT reactions allows for controlled growth of fibrous RP under low P$_4$ partial pressure. Second, the growth of fibrous RP along the radial direction is effectively inhibited by SnI$_2$ capping. The combination of these two effects yields high-quality fibrous RP flakes. Additionally, comprehensive polarized optical characterizations were conducted to thoroughly demonstrate the in-plane optical anisotropy of fibrous RP flakes. The orientation-dependent Raman polarization of fibrous RP flakes was observed in ARPRS, demonstrating its independence from sample thickness and excitation wavelength. Polarization-resolved visible light spectroscopy confirmed the anisotropic optical contrast, reflectance, transmittance and

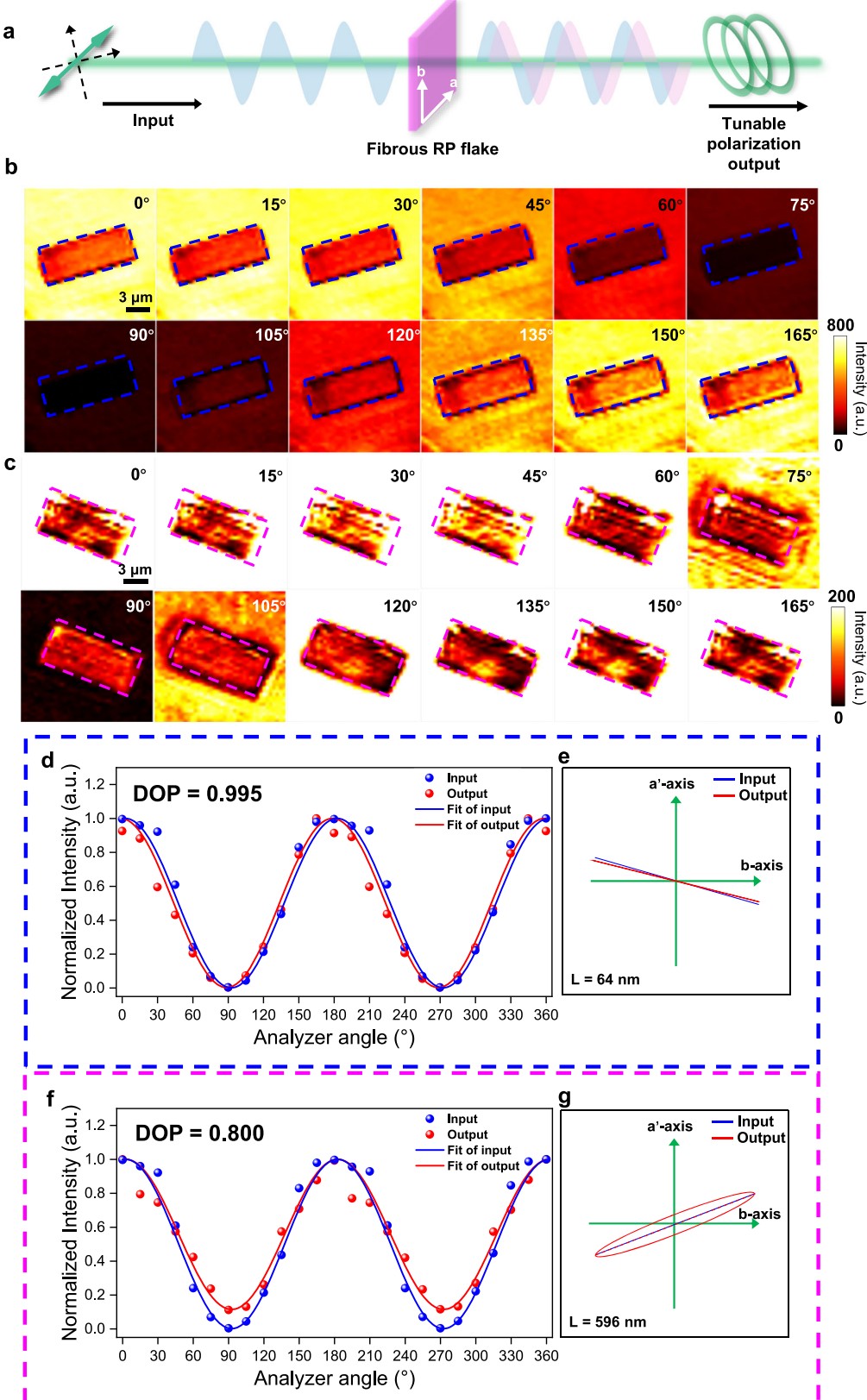

**Fig. 5 | A micro phase retarder based on fibrous RP flake. a** Schematic diagram of the birefringence in fibrous RP flake enabling polarization state conversion. **b, c** Polarization-resolved monochromatic transmitted light intensity mappings of the fibrous RP flakes at different analyzer angles (Analyzer angle from 0 to 165°, with a step size of 15°) with a thickness of 64 nm and 596 nm, respectively. The rectangular dashed box highlights the sample area. The color bar indicates the transmitted light intensity. **d, f** Polarization-dependent intensity of input (blue curve) spectra and output spectra (red curve) for the fibrous RP flakes corresponding to (**b**) and (**c**), respectively. Dots are experimental data, and the solid lines are the fitting curves. We define the degree of polarization (DOP) as $(I_{max} - I_{min})/(I_{max} + I_{min})$, where $I_{max}$, $I_{min}$ are the maximum light intensity and minimum light intensity, respectively. **e, g** Schematic diagram of polarization state for input/output light corresponding to (**d**) and (**f**), respectively.

absorption of fibrous RP flakes. Additionally, the combined use of PROM and DFT calculation revealed the presence of in-plane refractive index anisotropy in the flakes. Leveraging the significant in-plane refractive index anisotropy and substrate-independent growth behavior, we visually demonstrated the practical application of fibrous RP flakes as micro phase retarders for polarization conversion. By harnessing the distinctive characteristics of these flakes, we showcase their effectiveness in modulating and manipulating polarized light. Our work offers practical guidance for the two-dimensional growth of quasi-1D van der Waals materials, laying a solid foundation to further study the anisotropic properties of fibrous RP, providing a way for the construction of micro-optoelectronic devices in the future.

## Methods

### Growth of fibrous RP flakes on $SiO_2$/Si substrate

Fibrous RP flakes were directly grown on $SiO_2$/Si substrate by a chemical vapor transport reaction. Typically, RP (Alfa Aesar, 99.999%, ~400 mg), Sn powder (Alfa Aesar, 99.99%, ~20 mg), $I_2$ (Alfa Aesar, 99.9%, ~15 mg), and $SiO_2$/Si substrate were loaded into a quartz ampoule with an inner diameter of 10 mm. Afterward, the ampoule was evacuated to $10^{-3}$ Pa and sealed by vacuum sealing system (MRVS-1002, Partulab). A neck was introduced to the ampoule to separate the growth substrate from the source powders, as schematically shown in Fig. 1a. The ampoule was then placed horizontally in a tube furnace with the source zone at the low-temperature side and growth zone at the high-temperature side (Supplementary Fig. 1a). The zone, where the precursors were placed, was named source zone, and the zone, where the $SiO_2$/Si substrate was located and fibrous RP flakes were grown, was named growth zone. The furnace was heated to 620 °C for the growth zone (610 °C for the source zone) at a heating rate of 4 °C/min and maintained for 2 h, followed by cooling down to room temperature over 7 h. A detailed temperature program is presented in Supplementary Fig. 1b. The as-prepared samples, including bulk BP in the source zone and fibrous RP flakes on $SiO_2$/Si substrate in the growth zone (Supplementary Fig. 1c), were taken out by breaking the ampoule. Various growth substrates, including quartz (0001), Si (111), GaN (0001), sapphire (0001), mica, and soda-lime glass, were also chosen to explore the growth behavior of fibrous RP flake.

### Sample characterizations

**Optical micrograph and atomic force microscope (AFM).** The optical images were captured with a microscope equipped with a 50 times magnification optical objective (BX 51M, Olympus). The morphology of as-obtained fibrous RP flakes on the $SiO_2$/Si wafer was examined with AFM in tapping mode (Bruker Dimension Icon).

**Scanning electron microscope (SEM) and X-ray photoelectron spectroscopy (XPS).** SEM images were recorded by SEM (Hitachi SU8010) at 15 kV equipped with an energy-dispersive X-ray spectroscope (EDX). The valences of C, Si, and P elements were analyzed on an X-ray photoelectron spectrometer (PHI Quantera) using monochromatic Al K$\alpha$ ($h\nu$ = 1486.7 eV) as the excitation source.

**X-ray diffraction (XRD), electron backscattered diffraction (EBSD), and scanning transmission electron microscope (STEM).** The powder XRD patterns were measured on a Bruker D8 Advance diffractometer with Cu-K$\alpha$ radiation ($\lambda$ = 1.5406 Å) at 40 kV and 40 mA. The corresponding measurement configuration is schematically shown in the inset of Fig. 1h. Crystallinity and orientation of fibrous RP flakes were identified by an SEM equipped with an EBSD detector (TESCAN S9000X). During EBSD signals collection, the tilt angle was 70°, and the electron voltage was 20 kV with a collection speed of 114.2 Hz. The STEM image was obtained using an FEI Titan 80-300 operating at 300 kV. Lattice fringe images were captured by high-resolution TEM (JEOL JEM 2100F) at 200 kV.

**Time-of-flight secondary ion mass spectrometry (ToF-SIMS).** The ToF-SIMS measurements of fibrous RP flakes were performed by an SEM equipped with a SIMS detector (TESCAN S9000X). The analysis area was $60 \times 60$ μm$^2$, and the primary beam current and beam energy was 1.182 nA and 30 kV, respectively. During the ToF-SIMS signal collection, the ion mode was positive.

**Angle-resolved polarized Raman spectroscopy (ARPRS).** Raman spectrum was collected with a high-resolution confocal Raman system (Horiba Jobin Yvon LabRAM HR Evolution) using 532 nm laser excitation at room temperature. Raman mapping image was taken with a step of 0.2 μm. The angle-resolved polarized Raman spectra were measured under parallel- and cross-polarization configuration equipped with 514-, 532- and 633-nm laser sources. Collection time for every angle was 30 s at a laser power of about 60 μW.

**Angle-resolved polarized PL spectroscopy.** PL spectrum was collected by multifunctional spectral imaging microscope (Alpha300RAS, WITec) using 532 nm laser excitation at room temperature. PL mapping image was taken with a step of 0.2 μm. The angle-resolved polarized PL spectra were measured under parallel- and cross-polarization configuration equipped with 532 nm laser sources. Collection time for every angle was 1 s with 2 accumulations at a laser power of about 500 μW.

**Polarization-resolved visible light spectroscopy.** The polarization-resolved optical images were captured by altering the direction of the incident light from 0 to 180° with a step size of 15° by a multifunctional spectral imaging microscope (Alpha300RAS, WITec) equipped with a 100 times magnification optical objective (Fig. 4f). The polarization-resolved optical contrast spectrum, reflection spectrum and transmission spectrum were collected using white light excitation at room temperature. The collection time for every angle was 2 s with 4 accumulations.

**Polarization-resolved UV-vis-NIR absorption spectroscopy.** The polarized micro-area UV-vis-NIR absorption spectra were collected by a confocal spectrometer in the transmission mode (MStarter ABS, Matatest). The beam spot was about 2 μm. To eliminate the absorption of substrate, the fibrous RP flakes directly grown on transparent mica substrate were selected and analyzed.

**Polarization-resolved optical microscopy.** The polarization-resolved optical images were captured by a polarization microscope (LWT300) equipped with a 50 times magnification optical objective. The incident linearly polarized light was perpendicular to the analyzer in the transmission mode. The transmitted optical images were collected by rotating the sample from 0 to 180° with a step size of 15°.

**Polarization-resolved monochromatic transmitted light intensity spectroscopy.** The polarization-resolved monochromatic transmitted light intensity spectrum was collected by multifunctional spectral imaging microscope (Alpha300RAS, WITec) in the transmission mode with 532 nm laser excitation at room temperature. In the process of the measurement, the direction of the incident linearly polarized light was parallel to the x-axis of the laboratory coordinates, and the transmitted light intensity at 532 nm was collected by altering the analyzer direction from 0 to 180° with a step size of 15°. Collection time for every angle was 0.1 s with 20 accumulations at a laser power of about 1 μW. The polarization-resolved monochromatic transmitted light intensity mappings were taken with a step of 0.5 μm.

### Preparation of fibrous RP samples for TEM analysis

**PMMA-assisted wet transfer.** To identify the crystallographic orientation of a complete fibrous RP flake by TEM, a poly-methyl

methacrylate (PMMA)-assisted wet transfer technique was employed. First, the PMMA solution was spin-coated onto the $SiO_2$/Si substrate covered with fibrous RP flakes at 2500 rpm, followed by drying for 1 h. Afterward, the PMMA-coated substrate was immersed in a concentrated KOH solution to peel off the PMMA film together with underlying fibrous RP flakes. After that, the PMMA film was washed with DI water prior to being attached to the copper mesh. Finally, the PMMA film was dissolved in acetone to release the RP flakes to the copper mesh. The corresponding TEM results are shown in Fig. 2a–c.

**Focused ion beam (FIB) for cross-sectional STEM sample.** Atomic-resolution cross-sectional STEM sample was prepared by FIB (FEI Helios G4). First, a 5-µm-thickness Pt strap was deposited on the chosen location of samples in a specific direction which is perpendicular to the long parallel sides of the fibrous RP flake (Supplementary Fig. 12). The Pt strap prevents the fibrous RP sample from being damaged by the $Ga^+$ beam as well as provides mechanical support. The whole chosen sample, after milling, was lifted out and transferred to a copper half grid by an Easylift nanomanipulator. Then in the thinning stage, the sample was milled using a 30 kV $Ga^+$ beam with a current of 430, 230, 80 and 40 pA in turn. Finally, the possible damage area was removed using a 2 kV $Ga^+$ beam with a current of 15 pA for 1 min. The corresponding TEM results are shown in Fig. 2d–j.

### Theoretical calculations

**Gas phase species calculation.** The gas phase composition was evaluated with the TRAnsport–Gleichgewichten durch MINimierung der freien enthalpie (TRAGMIN) 5.1 code[46]. Thermodynamic data were set as default, if possible, or collected from *Thermochemical Data of Elements and Compounds*[47]. Seven gas phase species ($P_4$, $P_2$, $SnI_4$, $SnI_2$, $I_2$, I, Ar) and seven solid phases (Sn(s), P(s), $SnI_4$(s), $SnI_2$(s), Sn(l), $SnI_4$(l), $SnI_2$(l)) derived from our experiments were used as input parameters. Note that an inert gas specie is of necessity for the calculation setup, although in our actual experimental conditions, inert gas is not a must. We selected 10 mmol Argon in this case, while phosphorus, tin and iodine sources were set equivalent to 400, 15 and 10 mg to simulate experimental conditions. A simple one-room model with temperature series was built up to model CVT, with vessel volume set as 7.4 mL and temperature series as from 273 to 873 K.

**Growth mechanism analysis of fibrous RP flakes.** For the ab initio calculations on the intermediates toward the fibrous RP, the Gaussian16 code was employed with its default settings[48]. Structural optimizations and frequency calculations are carried out on the general gradient approximation (GGA) level with the Perdew-Burke-Ernzerhof functional (PBE)[49]. For the basis sets, the Gaussian16 integrated SDD setting was chosen, in which the Dunning full double zeta basis set[50] was applied for phosphorus, while Stuttgart/Dresden effective core pseudopotential (ECP) basis sets[51] were employed for tin and iodine. All structure models were visualized with the Visualization for Electronic and STructural Analysis program (VESTA)[52].

**Surface energy calculation.** Vienna ab initio simulation package (VASP) was used to perform first-principles calculations with density functional theory and the projector augmented wave (PAW) method[53,54]. The generalized gradient approximation (GGA) of the Perdew-Burke-Ernzerhof (PBE) standard functional was adopted for the exchange-correlation potential[49]. The cutoff energy for plane-wave expansion was set at 500 eV. The DFT-D3 method was used to describe the van der Waals interaction throughout the calculations[55]. Supercells expanded from 3 unit cells of fibrous red phosphorus with a vacuum region of 15 Å were applied to simulate the surface structure along the $a$, $b$, and $c$-axes, respectively. The surface geometry optimization started with the ideal atomic arrangement of the bulk fibrous red

phosphorus. Then in the supercells, one-third of the P atoms, which are closest to the surface, were allowed to relax. The total energy and the total force acting on each atom converged to $10^{-5}$ and 0.03 eV $Å^{-1}$. The $4 \times 4 \times 7$ and $1 \times 2 \times 4$, $2 \times 1 \times 4$, $2 \times 2 \times 1$ gamma-centered $k$-point meshes were respectively applied to the bulk model and supercells expanded along the $a$, $b$, and $c$-axes. The surface free energy is defined as

$$F = \frac{1}{2A}\left(E_{unrelax} - nE_{bulk}\right) + \frac{1}{A}\left(E_{relax} - E_{unrelax}\right), \qquad (7)$$

where $E_{unrelax}$ and $E_{relax}$ are the total energy of the unrelaxed and relaxed slab; $A$ is the surface area of the slab; $E_{bulk}$ is the energy of one P atom in the bulk; $n$ is the number of P atoms in the slab model. As is shown in the formula, surface free energy is acquired as the sum of two parts of energy relating to chemical bond breaking and surface relaxation.

**Optical property calculation.** Dielectric function calculation was conducted to determine the refractive index and optical absorption coefficient. For the optical property calculation, a denser gamma-centered $k$-point mesh of $6 \times 6 \times 11$ and a more precise energy tolerance of $10^{-7}$ eV were employed to assure the requirement of computational accuracy. The number of involved occupied valence and unoccupied conduction bands was raised to 360 for the determination of the frequency-dependent dielectric function under the independent particle approximation. With the dielectric function described by $\epsilon_1$ and $\epsilon_2$ as its real and imaginary part, the refractive index $n$ and absorption coefficient $\alpha$ can then be derived from the following equation:

$$\begin{cases} n = \left(\frac{\sqrt{\epsilon_1^2 + \epsilon_2^2} + \epsilon_1}{2}\right)^{1/2} \\ \kappa = \left(\frac{\sqrt{\epsilon_1^2 + \epsilon_2^2} - \epsilon_1}{2}\right)^{1/2}, \\ \alpha = \frac{2\omega\kappa}{c} \end{cases} \qquad (8)$$

where $\omega$ is the angular frequency of incident light, $c$ is the velocity of light in free space, $\kappa$ is the extinction coefficient.

## Data availability
All data supporting the findings of this study are available within the article and the Supplementary Information file. All raw data generated during the current study are available from the corresponding authors upon request.

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

## Acknowledgements

This work is supported by the National Natural Science Foundation of China (No. 21671115 and No. 52072198). We acknowledge the Tsinghua Xuetang Talents Program for providing computational resources. We cordially thank Prof. Tom Nilges and Dr. Markus R. P. Pielmeier at the Technical University of Munich for helpful discussions on the Gaussian calculation part of this research.

## Author contributions

Z.S. conducted the growth and structural characterizations of fibrous RP flakes. W.C. completed the exploration of growth mechanism, in-plane optical anisotropy measurement and displayed optical device applications of fibrous RP flakes. Z.S. and W.C. co-wrote the manuscript. B.Z. performed the theoretical calculations. Z.S., L.G., K.T. and Q.L. conducted ARPRS measurement. J.S. provided the optical test equipment and gave some guidance about experiments. Q.Y. conceived and supervised the project. All authors participated in the discussion and editing of the manuscript.

## Competing interests

The authors declare no competing interests.
