## [Peer Review File · Nature Communications]

Polarization Conversion in Bottom-up Grown Quasi-1D Fibrous Red Phosphorus FlakesREVIEWER COMMENTS

Reviewer #1 (Remarks to the Author):

This manuscript has reported the bottom-up growth of fibrous red phosphorus flakes and its application as micro phase retarders in polarization conversion. The RP flakes can be grown on diverse substrates with (001) preferred orientation, for which the authors demonstrate the KPFM and ARPRS characterization as well as the potential application for micro phase retarders. This paper is informative and well written. However, there are some concerns about the growth of the RP flakes.

1. I have noticed that the growth of RP nanowire on substrate has been realized (<https://onlinelibrary.wiley.com/doi/10.1002/anie.201605516>). Compared to these existing works on RP growth, the highlight of this paper should be the achievement on the growth of fibrous RP flakes on substrate. Can the authors demonstrate why the growth method in this work achieve the growth of RP flakes rather than common nanowire? The relative mechanism is unclear. And, are all the RP flakes grown in (001) orientation? There appears to be some inclination in the grown RP flakes as shown in in Fig.3.
2. The grown RP flakes have a lateral size of about 10 μm and thickness larger than tens of nanometers in this work. The grown flakes on diverse substrates shown in Fig.3 seem really thick, close to bulk crystal. Can the authors improve the growth controllability to obtain flakes with larger lateral size and smaller thickness? In my opinion, this produced RP flakes can hardly possess advantages over the flakes exfoliated from bulk.
3. In Fig.4 and corresponding statement, the authors demonstrate several characterizations to report the work function and anisotropy. However, I think the work function has no correlation with the later mainly demonstrated application about micro phase retarders. Besides, the anisotropy properties of RP has been studied by similar ARPRS and polarized PL in previous report (<https://www.nature.com/articles/s41467-021-25104-6>). In consideration of the prominence of the topic, it is better to place some supplementary contents in this part.

Overall, this work is valuable in the research field of RP materials. But I feel this manuscript is insufficient for its publication in Nature Communications.

Reviewer #2 (Remarks to the Author):

The authors demonstrated a bottom-up approach for the growth of highly crystalline RP flakes with (001)-preferred orientation on diverse substrates using an inverse-temperature-gradient-assisted CVT reaction. The work function, phonon vibrational anisotropy and in-plane refractive index anisotropy of RP flakes were explored, and their application as micro phase retarders in polarization conversion was visually demonstrated. However, the novelty of the study and the strength of the present results are still open to question. I do not recommend that this work be published in Nature Communication in its current form. The following issues should be addressed by the authors.

1. During the CVT process, will the introduction of Sn and I2 induce doping or contamination in the final RP flakes? The cross-section elemental mapping results in Fig. 2 should further include the data related to Sn and I.
2. The authors introduce their two-dimensional RP flakes in the manuscript, but in characteristics research and device application, the thickness of the used sample is hundreds of nanometers. In fact, these samples can hardly be called two-dimensional flakes. In Supplementary Fig. 6b, the authors show that they were able to obtain samples with a thickness of 20-80 nm. Follow-up studies should focus on these samples, which would be more valuable.
3. The Raman intensity at 390 cm^{-1} was weaker to be detected under cross polarization configuration than that under parallel polarization configuration, referring to Fig. 4e-f and Supplementary Fig. 14-28. Please explain the reason. In addition, the corresponding results should be presented in Supplementary Fig. 15, 18, 21, 24, 26, 28.
4. It is necessary to perform multiple mapping studies of specific Raman peaks or PL peaks for a whole flake. Raman and PL mapping results (images) are significant in studying the uniformity,

homogeneity, and stability of as-prepared samples.

5. Fixing the polarization direction of the incident light while changing the probe direction of the analyzer is not a sound way for angle-resolution polarized PL characterization. From the results presented by the authors, it can be found that when the angle of the analyzer changes, the polarization-dependent intensity of the substrate signal will also change. Test conditions, such as detector position and angle, should remain constant during measurement.

6. The work described in the manuscript mainly focuses on the preparation of RP flakes and corresponding devices, and these contents, including physical properties and device performance, are not directly related to BP. There is no need to add the introduction and data about BP (such as Supplementary Fig. 2) to the manuscript.

Reviewer #3 (Remarks to the Author):

The manuscript entitled "Polarization Conversion in Bottom-up Grown Quasi-1D Fibrous Red Phosphorus Flakes" presents the bottom-up approach for the growth of fibrous red phosphorus flakes via an inverse-temperature-gradient-assisted chemical vapor transport reaction. Highly crystalline red phosphorus flakes are directly grown on diverse substrates, which facilitates the explorations of the intrinsic properties of fibrous red phosphorus such as the work function, strong phonon vibrational anisotropy and in-plane refractive index anisotropy. The authors also demonstrate the application of fibrous red phosphorus flakes as microphase retarders in polarization conversion.

The manuscript is well and consistently written, and all the results are clearly formulated. The particular advantage of the manuscript is creating technology for obtaining high-quality red phosphorus flakes without transfer and exfoliation procedures being involved.

Cons/Questions:

1) I did not notice the citation of any significant works on strong (and even giant) anisotropy. In fact, there are no references at all to anisotropic works on the research topic (including quasi-one-dimensional materials); Do the authors really believe that no one has done any research in this area before them? I'm afraid that's absolutely unacceptable and should be fixed! Here I give only some recent works, which may be useful to read (and cite in the manuscript): Niu, S. et al. Giant optical anisotropy in a quasi-one-dimensional crystal. *Nat. Photonics* 12, 392–396 (2018). Segura, A. et al. Natural optical anisotropy of h-BN: highest giant birefringence in a bulk crystal through the mid-infrared to ultraviolet range. *Phys. Rev. Mater.* 2, 1–6 (2018). Ermolaev, G. et al. Giant optical anisotropy in transition metal dichalcogenides for next-generation photonics. *Nat Commun* 12, 854 (2021).

2) The use of eq. (3) for the description of polarization retardation ignores the reflection at the interfaces. Given that the refractive index is so high that the Fabry-Perot peaks emerge in PL spectra and the giant anisotropy the eq. (3) should involve transmittance amplitudes for a planar multilayer structure. At present, the interpretation gives wrong estimates of k . Also, data on DOP (p. 15) are left uninterpreted, likely due to the same reason. In particular, it is interesting to understand the reason why DOP at 681 and 90-nm-thick flakes are so low.

3) Does the spacing of Fabry-Perot peaks agree with other data on in-plane anisotropy like DFT calculations of n_1 , n_2 , and phase retardation estimations?

4) It is mentioned that the reported in-plane anisotropy is greater than that of As₂S₃, but it seems to be high in the high absorption region, which the authors do not explicitly state. There is also another nuance: for some reason, authors do not provide experimental data in the spectral range but show only DFT results. The authors do not seem to extract any constants from the polarization transmission spectra. Instead, they pointed out that extracting the refractive index for specific wavelengths is possible using photoluminescence spectra and even wrote a formula. Still, I can not see any numbers or explicit indications of experimental data (only DFT is given).

5) Side remark: I noticed multiple uses of "angle-resolution polarized photoluminescence", shouldn't it be "angle-resolved"?

Point-by-Point Responses to the Reviewers' Comments

Reviewer # 1

1. I have noticed that the growth of RP nanowire on substrate has been realized (<https://onlinelibrary.wiley.com/doi/10.1002/anie.201605516>). Compared to these existing works on RP growth, the highlight of this paper should be the achievement on the growth of fibrous RP flakes on substrate. Can the authors demonstrate why the growth method in this work achieve the growth of RP flakes rather than common nanowire? The relative mechanism is unclear. And, are all the RP flakes grown in (001) orientation? There appears to be some inclination in the grown RP flakes as shown in in Fig.3.

Response 1: We would like to express our gratitude to the reviewer for bringing the important issue to our attention, i.e., the growth mechanism of the fibrous RP flakes. After in-depth experimental and theoretical study, we now conclude that the formation of fibrous RP flaks is attributed to the Sn-mediated low P₄ partial pressure and the SnI₂ capping layer-directed growth of fibrous RP.

First of all, the growth rate of each crystal facet is intricately linked to the interplay between the chemical potential difference ($\Delta\mu$) and kinetic barrier (ΔG) [Ref. 27: *Nature*, **374**, 342-345 (1995), Ref. 28: *Ind. Eng. Chem. Res.* **47**, 9812–9833 (2008), Ref. 29: *Chem. Commun.* **50**, 4620-4623 (2014), Ref. 30: *Nat. Commun.* **10**, 761 (2019)]. Generally, ΔG is inversely proportional to the surface energy (γ) [Ref. 29: *Chem. Commun.* **50**, 4620-4623 (2014), Ref. 30: *Nat. Commun.* **10**, 761 (2019), Ref. 31: *Adv. Mater.* **15**, 441 (2003)]. According to the surface energy of fibrous RP calculated by DFT (Fig. R1, Left), $\gamma_{(010)} \gg \gamma_{(100)} > \gamma_{(001)}$. Consequently, $\Delta G_{(010)} \ll \Delta G_{(100)} < \Delta G_{(001)}$ (Fig. R1, Right). The chemical potential difference can be modified by manipulating the synthesis temperature or controlling the partial pressure of gas phase species, thereby influencing the diverse morphologies observed in crystal growth. The element Te, which has a similar quasi-one-dimensional (quasi-1D) structure, exhibits diverse morphologies, such as 1D rod, 2D flake and 3D sphere, under various synthesis temperatures [Ref. 32: *iScience* **25**, 103594 (2022)]. Differently, the partial pressure of gas phase species plays a crucial role in controlling the morphology of fibrous RP. In the P/Sn/I₂ system, the addition of tin allows black phosphorus to gain an advantage in competition with fibrous RP, consuming most of the available phosphorus source [Ref. 35: *J. Phys. Chem. Lett.* **9**, 1759-1764 (2018)]. Additionally, an inverse temperature gradient was adopted to hinder the transfer of P₄ molecules to the substrate, leading to a further reduction in the partial pressures of P₄ molecules at the growth zone. Therefore, the Sn in the P/Sn/I₂ system can manipulate the P₄ partial pressure in the growth zone, fascinating the formation of fibrous RP in our CVT reaction.

Fig. R1 Left: Surface energy of (100), (010), and (001) planes for fibrous RP.

Right: Schematic diagram of kinetic barrier of each crystal face.

When the P_4 partial pressure maintains very low, P_4 molecules tend to grow along the longitudinal direction, resulting in the formation of fibrous RP nanowires or nanorods (Fig. R2, i); with a slight increase in P_4 partial pressure, lateral growth becomes favorable, leading to the formation of fibrous RP flakes (Fig. R2, ii); when P_4 partial pressure further increases, isotropic rapid growth occurs, and fibrous RP polycrystalline spheres are formed (Fig. R2, iii). It is worth mentioning that under low P_4 partial pressure in the P/Sn/I₂ system, fibrous RP 1D rods (few) and flakes (most) are observed simultaneously (Figs. R3a, b). However, in the P/I₂ system without competitive consumption of most P_4 molecules through the growth of black phosphorus, only bulk fibrous RP polycrystalline spheres are formed owing to the high P_4 partial pressure presented at the growth zone (Fig. R3c).

Fig. R2 Schematic illustration for the morphology evolution of fibrous according to the relationship between chemical potential difference and kinetic barrier.

Fig. R3 Morphology evolution of fibrous RP under low/high P_4 partial pressure in the P/Sn/I₂ and P/I₂ system, respectively.

Fig. R4 Partial pressure diagram of the corresponding equilibrium gas pressure of P/Sn/I₂ system in the temperature ranges from 273 K to 873 K calculated by CalPhaD.

Meanwhile, according to the calculations of gas phase species, the partial pressure of P₄ and SnI₂ at equilibrium is at least two orders of magnitude higher than the competitive molecules (i.e. SnI₄ and I₂), suggesting that SnI₂ may contribute to the growth of fibrous RP flakes (Fig. R4). We employed Gaussian to calculate the stable configuration of fibrous RP intermediates, using the energy of starting materials as the reference. Due to the unique bonding configuration of fibrous RP, there is likely an "addition–elimination" equilibrium of SnI₂ molecules during its growth along the tube. Similar to the formation of black phosphorus [Ref. 36: *Angew. Chem. Int. Ed.* **60**, 6816–6823 (2021)], Sn atoms in SnI₂ may coordinate with phosphorus atoms at the edge of the parallel twin tubes. The coordination of double [P9] subunit with four SnI₂ entities to **i** (Fig. R5a) is electronically favored by total energy difference $\Delta E_{\text{tot}} = -658$ kJ/mol. Then, in order to lower the energy of the system, this bond between phosphorus and tin will be broken and reformed, allowing addition of P₄ molecules along the tube to yield **ii** (Fig. R5a). We also calculated the intermediate of fibrous RP without SnI₂ capping (Fig. R5a, **iii**). In this premise, fibrous RP is expected to grow spontaneously with successive addition of P₄ atoms, leading to the growing preference along the b-axis. Yet it is not the case. The energy of this configuration is slightly higher by 319 kJ/mol compared to the SnI₂-capped structure, indicating that the capping of SnI₂ is energetically advantageous and more promising in P/Sn/I system (Fig. R5b). Since the SnI₂ capping layer significantly stabilizes fibrous RP and inhibits its growth along the b-axis, it further promotes the growth of parallel phosphorus tubes along the direction of van der Waals forces, favoring the formation of two-dimensional (2D) flakes with low aspect ratio rather than one-dimensional rods/wires (1D) with high aspect ratio.

Fig. R5 a Schematic diagram of the starting materials (P₄, SnI₂) and geometry-optimized structures (**i**, **ii**, **iii**) in the proposed phosphorene mechanism. **m** Total Energy of the starting materials (P₄, SnI₂) and geometry-optimized structures (**i**, **ii**, **iii**) calculated by Gaussian.

Fig. R6 **a** SEM images of the selected area. **b** Mass spectra of tin and iodine. **c** Depth profile of the P^+ , $[PO]^+$, $[P_xO_yH_z]^M+$, Sn^+ and I^+ within the selected area. Inset: an enlargement of the red dotted rectangle. The surface and cross-sectional ion analysis mapping of **d** P^+ , **e** $[PO]^+$, **f** $[P_xO_yH_z]^M+$, **g** Sn^+ and **h** I^+ measured by ToF-SIMS. Top: surface ion analysis mapping. Down: cross-sectional ion analysis mapping.

To check whether the SnI_2 capping layer still exists or not after the formation of fibrous RP flakes, we conducted a detailed analysis of the element distribution about the depth of the fibrous RP surface using time-of-flight secondary ion mass spectrometry (ToF-SIMS). Our results indicate that there were no detectable I^+ signal and an extremely weak Sn^+ signal in the selected ion bombardment area. This is because, during the cooling process, SnI_2 undergoes decomposition into Sn and I_2 again. While the former remains on the substrate surface, the latter diffuses randomly and can exist at any position within the ampoule. Specifically, we observed five peaks that correspond to five isotopes of tin, namely ^{116}Sn , ^{117}Sn , ^{118}Sn , ^{119}Sn , and ^{120}Sn (Figs. R6a, b). It is worth noting that fibrous RP often exists as high mass-to-charge ratio phosphorus cluster fragments after being bombarded [Ref. 37: *Chem. Eng. J.* **421**, 127841 (2021)], which are beyond the detection range of the device. Consequently, the signal of low mass-to-charge ratio phosphorus ion is weak (Figs. R6c, d, e, f). Furthermore, the depth distribution image revealed that Sn was only present on the substrate surface, and there was no signal distribution for detecting Sn or I inside the sample (Figs. R6c, g, h).

Fig. R7 **a** Partial pressure diagram of the corresponding equilibrium gas pressure of P/I_2 system in the temperature ranges from 273 K to 873 K calculated by CalPhaD **b** Schematic diagram of the geometry-

optimized structures of fibrous RP capped with single iodine atom.

Fig. R8 Optical images of the fibrous RP flakes with high aspect ratio obtained in the improved P/I₂ system.

We also checked the feasibility of I₂ as a capping layer. As indicated by the CalPhaD calculations, in the absence of Sn, the main gas phase species with the most significant partial pressure are P₄ and I₂ (Fig. R7a). Although the structure capped with iodine also converges, it is bonded to the edge in the form of a single iodine atom (Fig. R7b), which is clearly not the main species in the gas phase. Moreover, the structure capped with iodine deviates from the original configuration of fibrous RP. Thus, we can conclude that in this scenario, iodine serves solely as a traditional CVT transport agent and does not have any capping effect. Experimentally, we created a low partial pressure region of P₄ in the P/I₂ system under the same temperature conditions by reducing the amount of red phosphorus and controlling mass transfer. As anticipated, we were able to obtain stacked, high aspect ratio fibrous RP flakes that were hundreds of microns long under low P₄ pressure without the presence of Sn (Fig. R8). However, it should be noted that the crystal quality, orientation, and thickness of these flakes are not comparable to the fibrous RP flakes with low aspect ratio that are capped with SnI₂.

Fig. R9 Schematic diagram of the growth mechanism of fibrous RP crystals.

Based on the above discussions, the answer to the question “*Why the growth method in this work achieve the growth of RP flakes rather than common nanowire*” becomes clear. Fibrous RP exhibits diverse morphologies depending on the variations in P₄ partial pressure. In the P/Sn/I₂ system, Sn plays a crucial role in regulating the partial pressure at the growth zone in the CVT reaction. Simultaneously,

the capping effect of SnI₂ may direct the 2D growth behavior of fibrous RP. The synergistic effect of Sn-mediated P₄ partial pressure and the SnI₂ capping layer-directed growth facilitates the formation of fibrous RP flakes (Fig. R9).

To address the concern regarding the growth mechanism in this comment, all the above discussion has been added to the section “**Mechanism for the Growth of Fibrous RP Flakes**” in the revised manuscript with revision traces, see Lines 245-360 from Page 10 to Page 15.

In addition, Fig. R1, Fig. R2 (Right), Fig. R3, Fig. R5 and Fig. R6 have been incorporated into Fig. 3 in the revised manuscript. Fig. R2 (Left), Fig. R4, Figs. R7-R9, have been added to the revised supplementary information as Supplementary Table 1, Supplementary Figs. 13-16, respectively.

Response 2: We appreciate the reviewer's insightful question regarding the orientation of the RP flakes. In our study, we conducted Powder X-ray diffraction (PXRD) analysis on the fibrous RP directly grown on SiO₂/Si substrate. Fig. R10a displays two distinct diffraction peaks, specifically indexed as (001) and (002) planes, confirming that the majority of the fibrous RP flakes exhibit a preferred (001) growth orientation.

Indeed, during the growth of fibrous RP flakes, local fluctuations in the supply of gas phase species might cause the vertical thickening and step formation of the fibrous RP, resulting in some inclined flakes (Fig. R10a, pink circle remark). Nevertheless, EBSD measurement on this sample, similar to those depicted in Fig. 3 of the original manuscript, indicates that the exposed surface at the top remains the (001) surface (Fig. R10b).

Fig. R10 **a** SEM image of the fibrous RP flakes grown on the SiO₂/Si substrate. **b** The corresponding inverse pole figure (IPF) along the z-axis, demonstrating the distinct (001) crystal orientation.

To clarify, we have added “*Specifically, during the growth of fibrous RP flakes, perturbation of gas phase species supply leads to the formation of steps in the vertical direction, which may appear inclined at certain angles. Nevertheless, the top exposed surface consistently maintained a preferred orientation of (001), as depicted in Supplementary Fig. 7.*” in the revised manuscript with revision traces, see Lines 184-187 on Page 7.

In addition, Figs. R10b and R10c have been incorporated into the revised supplementary information as Supplementary Fig. 7.

2. The grown RP flakes have a lateral size of about 10 μm and thickness larger than tens of nanometers in this work. The grown flakes on diverse substrates shown in Fig.3 seem really thick, close to bulk crystal. Can the authors improve the growth controllability to obtain flakes with larger lateral size and smaller thickness? In my opinion, this produced RP flakes can hardly possess advantages over the flakes exfoliated from bulk.

Response 3: We really appreciate the reviewer's kind suggestion. According to the proposed growth mechanism in the response to Comment 1, the formation of fibrous RP flakes at the growth zone relies on two factors, i.e., the appropriate consumption of phosphorus sources by black phosphorus generation at the source zone and the capping effect of SnI_2 . The uncontrolled consumption of the phosphorus source poses a significant challenge to achieving precise control over the thickness of fibrous RP flakes. If the competitive growth of black phosphorus consumes almost all the phosphorus sources at the source zone, it restricts the crystal nucleation of fibrous RP, resulting in the absence of fibrous RP flakes on the substrate at the growth zone. Conversely, if the competitive growth of black phosphorus is insufficient to consume an adequate amount of phosphorus sources, and SnI_2 inhibits radial growth, the excess phosphorus source will promote the growth of fibrous RP in the thickness direction. This leads to the formation of thick flakes and may even result in the creation of growth steps (Fig. R11). Accordingly, it is difficult to directly manipulate the growth of fibrous RP flakes with larger sizes or thicknesses approaching fewer layers by simply adjusting the feeding or temperature parameters.

Fig. R11 SEM images of the thick fibrous RP flakes.

Additionally, when it comes to other phosphorus allotropes with two-dimensional layer structures, such as black phosphorus or violet phosphorus, they are held together by interlayer bonds through van der Waals forces. This characteristic allows for the possibility of obtaining 2D phosphorus flakes with only a few or even a single layer through mechanical or liquid phase exfoliation. However, for quasi-one-dimensional materials, van der Waals force acts in two directions. Consequently, only 1D rods or ribbons can be obtained through mechanical or liquid phase exfoliation, accompanied by the presence of numerous surface defects and irregularities. As a result, previous attempts at exfoliating fibrous RP flakes from bulk crystals have yielded only poor-quality and defective samples (Fig. R12). We have previously undertaken similar endeavors, but have been unable to achieve fibrous RP flakes through the exfoliation process from bulk crystals (Fig. R13). All of the aforementioned results conclusively demonstrate the infeasibility of obtaining fibrous RP flakes directly exfoliated from bulk crystals. Therefore, we believe that the direct growth of fibrous RP flakes on the substrate holds significant importance and surely possesses advantages over the flakes exfoliated from bulk in the future study of fibrous RP.

Fig. R12 **a, b, c, d** TEM images of the fibrous RP samples obtained by liquid phase exfoliation from bulk crystals. **e, f** Optical images, **g** AFM image and **h** SEM image of the fibrous RP samples obtained by mechanical exfoliation from bulk crystals.

[Fig. R12 a, d: *J. Photochem. Photobiol. A.* **384**, 112034 (2019)]

[Fig. R12 b: *Chem. Mater.* **33**, 6240-6248 (2021)]

[Fig. R12 c: *J. Mater. Chem. A* **9**, 338-348 (2021)]

[Fig. R12 e: *Inorg. Chem.* **59**, 976–979 (2020)]

[Fig. R12 f, g: *Nat. Comm.* **12**, 4822 (2021)]

[Fig. R12 h: *Inorg. Chem.* **60**, 4883–4890 (2021)]

Fig. R13 **a** Optical images of the fibrous RP rods obtained by mechanical exfoliation from bulk crystals. **b** TEM images of the fibrous RP ribbons obtained by liquid phase exfoliation from bulk crystals.

3. In Fig.4 and corresponding statement, the authors demonstrate several characterizations to report the work function and anisotropy. However, I think the work function has no correlation with the later mainly demonstrated application about micro phase retarders. Besides, the anisotropy properties of RP has been studied by similar ARPRS and polarized PL in previous report (<https://www.nature.com/articles/s41467-021-25104-6>). In consideration of the prominence of the topic, it is better to place some supplementary contents in this part.

Response 4: We would like to express our sincere gratitude to Reviewer 1 for the valuable comments and insightful suggestions. In accordance with Reviewer 1's recommendation, we have removed the section discussing the work function from the revised manuscript, as it is not directly related to the overall content of the article. Additionally, considering the observation of the antenna effect for the first time in fibrous red phosphorus flakes and the completeness of the discussion on optical anisotropy, we have chosen to retain the relevant content of ARPRS in the revised manuscript. In order to streamline the main text, we have moved the section on angle-resolved polarized PL spectra to the Supplementary Information. This decision was made based on the similarity of this technique to ARPRS and the fact that it has been previously reported [Ref 5: *Nat. Comm.* **12**, 4822 (2021)].

Fig. R14 Optical images of fibrous RP irradiated by polarized white light.

Fig. R15 Polarization-resolved optical contrast spectra of **a** horizontally (I) and **b** vertically placed (II) fibrous RP flakes. **c, d** Polarization-resolved reflection spectrum and transmission spectrum of the fibrous RP flakes, respectively.

However, to establish a connection with the subsequent application of phase retarders, we have supplemented additional content in the revised manuscript that focuses on polarization-resolved visible light spectroscopy. These sections encompass optical contrast images and spectra (Fig. R14 and Figs. R15a, b), reflection spectra (Fig. R15c), transmission spectra (Fig. R15d), and polarization-resolved optical microscopy (Fig. R16). Together, they provide a comprehensive understanding of the in-plane optical anisotropy resulting from the refractive index anisotropy.

Fig. R16 Transmitted images of fibrous RP flakes under crossed polarized light.

To clarify, we have added “Moreover, polarization-resolved visible light spectroscopy was performed to investigate the in-plane optical anisotropy. To ensure PROM measurement can clearly display the in-plane anisotropic refraction of fibrous RP flakes.” in the section “**In-plane Optical Anisotropy of Fibrous RP Flakes**”, see Lines 487-528 from Page 22 to Page 23 in the revised manuscript with revision traces. In addition, Figs. R14-R16 have been incorporated into Fig. 4 in the revised manuscript.

Once again, we express our sincere appreciation to Reviewer 1 for the valuable recommendation, which has significantly contributed to the improvement and completeness of our manuscript.

Reviewer # 2

1. During the CVT process, will the introduction of Sn and I₂ induce doping or contamination in the final RP flakes? The cross-section elemental mapping results in Fig. 2 should further include the data related to Sn and I.

Response: We sincerely appreciate the valuable suggestion provided by the reviewer. In Fig. 2, the elemental mappings of the cross-sectional fibrous RP flake on the SiO₂/Si substrate were obtained using the Low-magnification High-Angle Annular Dark-Field Scanning Transmission Electron Microscopy (HAADF-STEM) technique. Regrettably, during that particular data collection, there were only noise signals present, and as a result, no element mapping of Sn and I was preserved.

Fig. R17 Cross-sectional SEM image of the fibrous RP flake and the corresponding elemental mappings of O, Si, P, Sn and I.

In order to address this concern, we employed the FIB technique to acquire a new cross-sectional TEM sample. Subsequently, we collected the elemental mappings of Sn and I using High-Resolution TEM. Fig. R17 illustrates that the signals corresponding to the elemental mappings of Sn and I exhibited consistency with the background noise. This outcome demonstrates that the presence of Sn and I within the fibrous RP flakes could not be detected due to the detection limit of High-Resolution TEM.

We also employed Time-of-Flight Secondary Ion Mass Spectrometry (ToF-SIMS) to investigate the presence of Sn or I in the fibrous RP flakes. Despite using a relatively large selected area of $60 \times 60 \mu\text{m}^2$, no detectable I^+ signal was observed, and there was only an extremely weak Sn^+ signal within the area, as depicted in Fig. R6b. As discussed in the growth mechanism section *“During the cooling process, SnI_2 undergoes decomposition into Sn and I_2 again. While the former remains on the substrate surface, the latter diffuses randomly and can exist at any position within the ampoule”*, the depth distribution image revealed that Sn was solely present on the substrate surface, and there was no signal distribution for detecting Sn or I inside the sample (Figs. R6 c, g, h). Hence, we believe that Sn and I_2 will not be introduced into the fibrous RP flakes as dopant or contaminant during the CVT process.

2. The authors introduce their two-dimensional RP flakes in the manuscript, but in characteristics research and device application, the thickness of the used sample is hundreds of nanometers. In fact, these samples can hardly be called two-dimensional flakes. In Supplementary Fig. 6b, the authors show that they were able to obtain samples with a thickness of 20-80 nm. Follow-up studies should focus on these samples, which would be more valuable.

Response: We sincerely thank the reviewer for their valuable suggestion. It is worth noting that the phase retarder value (δ) is highly dependent on the thickness (d) of the optical material (Equation 1).

$$\delta \equiv \frac{\Delta n 2\pi d}{\lambda}. \quad (1)$$

Therefore, if we were to solely select samples with thickness ranging from 20 to 80 nm to demonstrate their device application in micro phase retarder, the modulation of the polarization state of light would be minimal, and the resulting differences would not be significant. As shown in Fig. R18a, the fibrous RP with a thickness of 64 nm within the blue box exhibits consistent changes in the transmitted light intensity along with the substrate. This indicates that the intensity undergoes complete extinction at 90° , similar to the substrate behavior. For the thin sample (64 nm), the output spectrum closely matches the input spectrum, consistent with the mapping results (Fig. R18c). Additionally, the DOP value is 0.995, indicating a minimal change in the polarization state of the incident light (Fig. R18d). While for the fibrous RP with a thickness of 596 nm within the pink box, the transmitted light intensity maintains a consistent value despite reaching complete extinction at 90° (Fig. R18b). For the thick sample (596 nm) with a DOP of 0.8, the incident linearly polarized light is transformed into elliptically polarized light (Figs. R18f, g).

Fig. R18 a, b Polarization-resolved transmission spectral mappings of the fibrous RP flakes at different analyzer angles (Analyzer angle from 0° to 165° , with a step size of 15°) with a thickness of 64 nm and 596nm, respectively. Rectangular dashed box highlights the sample area. Colour bar indicates the transmitted light intensity. **c, e** Polarization-dependent intensity of input/output spectra for the fibrous RP flakes corresponding to Fig. 5a and Fig. 5b, respectively. **d, f** Schematic diagram of polarization state for input/output light corresponding to Fig. 5c and Fig. 5e, respectively.

To address the reviewer’s concern, with careful consideration of the above factors, we have chosen to include samples with a wide thickness range (64 nm, 100 nm, 147 nm, 178nm and 596 nm) in order to effectively present our device’s performance, see “*To demonstrate the polarization state conversion of incident light using fibrous RP flakes of varying thicknesses, we observed that the DOP of the transmitted light is 0.981, 0.916, and 0.873 for fibrous RP flakes with thicknesses of 100 nm, 147 nm, and 178 nm, respectively (Supplementary Figs. 41-43)*” in Line 663-666 on Page 29 in the revised

manuscript with revision traces. Fig. R18 was also incorporated into Fig. 5 in the revised manuscript and Additional experimental data were incorporated into Supplementary Figs. 41-43.

3. The Raman intensity at 390 cm^{-1} was weaker to be detected under cross polarization configuration than that under parallel polarization configuration, referring to Fig. 4e-f and Supplementary Fig. 14-28. Please explain the reason. In addition, the corresponding results should be presented in Supplementary Fig. 15, 18, 21, 24, 26, 28.

Response: We really appreciate the reviewer's kind suggestion. The Raman tensor of the fibrous RP can be expressed as:

$$R(A_g) = \begin{pmatrix} a & d & e \\ d & b & f \\ e & f & c \end{pmatrix}. \quad (2)$$

Normally, different materials exhibit distinct Raman vibration peaks, and these peaks correspond to different Raman tensors. Essentially, the Raman tensors consist of various elements arranged in a matrix.

Thus, the observed anisotropic Raman scattering intensity for A_g mode can be expressed by the following equations¹:

$$I(A_g, //) = (b \cos\theta + d \sin\theta)^2, \quad (3)$$

$$I(A_g, \perp) = (d \cos\theta + a \sin\theta)^2. \quad (4)$$

Specifically, referring to Equation 3 and Equation 4, if the Raman tensor matrix element a at 390 cm^{-1} is significantly smaller than b , then the Raman peak intensity at 390 cm^{-1} will be considerably weaker in the cross-configuration compared to the parallel-configuration. Moreover, the weak Raman peak intensity at 390 cm^{-1} contributes to the proximity of the signal noise to the variation in Raman peak intensity during the ARPRS measurement. Consequently, this proximity results in a substantial deviation from the fitting. This occurrence is not uncommon, and it can be observed, for instance, in the considerable variation of Raman peak intensities at 401 cm^{-1} and 275 cm^{-1} due to different configurations (Fig. R19).

Fig. R19 Contour plot of polarized Raman intensity under parallel (//) and cross (\perp) polarization configuration.

To address the reviewer's concern, we have added "However, the Raman vibration peak at 390 cm^{-1} exhibits a strong signal and significant change in the parallel-configuration, whereas the signal is weak and shows little change in the cross-configuration (Fig 4b, 4c and Supplementary Figs. 19, 22, 25, 28, 30, 32). This discrepancy can be attributed to the difference in the Raman tensor element of fibrous RP in the two configurations at 390 cm^{-1} .", see Line 434-439 on Page 20 in the revised manuscript with revision traces.

In addition, we have supplemented the polar plots for the Raman peaks at 390 cm^{-1} of the fibrous RP flake under cross-polarization configuration in the revised supplementary information as Supplementary Figs. 19, 22, 25, 28, 30, 32.

4. It is necessary to perform multiple mapping studies of specific Raman peaks or PL peaks for a whole flake. Raman and PL mapping results (images) are significant in studying the uniformity, homogeneity, and stability of as-prepared samples.

Response: We thank the reviewer for the valuable suggestion. Fig. R20a and Fig. R20b show the Raman spectrum and corresponding Raman mapping at 471 cm^{-1} ; Figs. R20c-20f show the PL spectrum and corresponding mapping of photoluminescence intensity, FWHM and peak, respectively. Both Raman and PL mappings results demonstrate the homogeneity and stability of as-prepared samples.

Fig. R20 **a** Raman spectrum of fibrous RP flakes measured under 532 nm excitation. The peak labeled “*” originates from the SiO_2/Si substrate. **b** Raman mapping at $\sim 471\text{ cm}^{-1}$ of the fibrous RP flake. **c** PL spectrum of the fibrous RP flake. **d-f** Mapping of photoluminescence intensity, FWHM and peak, respectively.

To clarify, we have added “*Raman mapping and photoluminescence (PL) mapping for the fibrous RP flake exhibit uniform signals across the whole flake, demonstrating high continuity and homogeneity of the as-prepared fibrous RP flake (Fig. 1e, inset, and Supplementary Fig. 5).*”, see Line 164-167 on Page 7 in the revised manuscript with revision traces.

In addition, Fig. R20b has been added into the revised manuscript as inset of Fig. 1e. Figs. R20c-20f have been added to the revised supplementary information as Supplementary Fig. 5.

5. Fixing the polarization direction of the incident light while changing the probe direction of the analyzer is not a sound way for angle-resolution polarized PL characterization. From the results presented by the authors, it can be found that when the angle of the analyzer changes, the polarization-dependent intensity of the substrate signal will also change. Test conditions, such as detector position and angle, should remain constant during measurement.

Response: We sincerely appreciate the valuable suggestion provided by the reviewer. In order to

eliminate any potential ambiguity, we conducted additional tests on the angle-resolved polarized PL spectra of fibrous RP flakes. The testing procedure involved consistent sample orientation, angle definition, and measurement configuration in accordance with the ARPRS method. Similar to the ARPRS results, we observed that the PL intensity of the fibrous RP flakes is orientation-dependent, with the peak intensity reaching its maximum at 90° and minimum at 0° and 180° angles. The slight variations in peak position under the two configurations are attributed to differences in spot position during testing. Moreover, we would like to acknowledge the suggestion made by Reviewer 2 in Comment 2, wherein we specifically selected thin samples for testing to eliminate the Fabry–Pérot effect caused by multi-beam reflection in thick samples.

Fig. R21 Contour plot of polarized PL intensity under **a** parallel ($//$) and **c** cross (\perp) polarization configuration. Polar plots of the polarized PL intensity under **b** parallel ($//$) and **d** cross (\perp) polarization configuration.

To address this comment, we have significantly simplified the discussion of the reported polarized PL of fibrous RP as it has been noted in previous report [Ref 5: Nat. Comm. 12, 4822 (2021)]. Moreover, we have omitted the relevant introduction and description of the Fabry–Pérot effect to ensure a more concise presentation in the revised manuscript, see Line 447-486 from Page 21 to Page 22 in the revised manuscript with revision traces.

In addition, Fig. R21 has been added into the revised supplementary information as Supplementary Fig. S33.

6. The work described in the manuscript mainly focuses on the preparation of RP flakes and corresponding devices, and these contents, including physical properties and device performance, are not directly related to BP. There is no need to add the introduction and data about BP (such as Supplementary Fig. 2) to the manuscript.

Response: We thank the reviewer for the comments. We agree with the reviewer’s suggestion that the content in the Introduction section concerning black phosphorus can be deleted as it is slightly related the research topic this article. Nevertheless, we found that the growth mechanism of fibrous RP flakes was closely related to black phosphorus during the study, as stated in the response to Comment 1 by Review 1.

Based on the suggestions of reviewers and the integrity of the article content, we deleted the description of black phosphorus in the Introduction section and retained the content of black phosphorus in the later growth mechanism section.

Once again, we would like to extend our heartfelt gratitude to Reviewer 2 for the valuable recommendation, which has greatly enhanced the quality and comprehensiveness of our manuscript.

Reviewer # 3

1. I did not notice the citation of any significant works on strong (and even giant) anisotropy. In fact, there are no references at all to anisotropic works on the research topic (including quasi-one-dimensional materials); Do the authors really believe that no one has done any research in this area before them? I'm afraid that's absolutely unacceptable and should be fixed! Here I give only some recent works, which may be useful to read (and cite in the manuscript): Niu, S. et al. Giant optical anisotropy in a quasi-one-dimensional crystal. *Nat. Photonics* 12, 392–396 (2018). Segura, A. et al. Natural optical anisotropy of h-BN: highest giant birefringence in a bulk crystal through the mid-infrared to ultraviolet range. *Phys. Rev. mater.* 2, 1–6 (2018). Ermolaev, G. et al. Giant optical anisotropy in transition metal dichalcogenides for next-generation photonics. *Nat Commun* 12, 854 (2021).

Response: We express our gratitude to the reviewer for bringing these significant prior works to our attention. In response, we have incorporated references to these works in the introduction section of the revised manuscript [Ref. 16: *Nat. Commun.* 12, 854 (2021), Ref. 17: *Phys. Rev. Mater.* 2, 024001 (2018) and Ref. 23: *Nat. Photonics* 12, 392-396 (2018)]. Furthermore, we have included citations to other notable studies that emphasize the significance of quasi-one-dimensional materials in relation to optical anisotropy [Ref. 18: *J. Am. Chem. Soc.* 138, 300-305 (2016), Ref. 19: *ACS Photonics* 4, 3023-3030 (2017), Ref. 20: *ACS Photonics* 5, 2509-2515 (2018), Ref. 21: *Nanoscale* 14, 12238 (2022) and Ref. 22: *Adv. Mater.* 29, 1700441 (2017)].

2. The use of eq. (3) for the description of polarization retardation ignores the reflection at the interfaces. Given that the refractive index is so high that the Fabry-Perot peaks emerge in PL spectra and the giant anisotropy the eq. (3) should involve transmittance amplitudes for a planar multilayer structure. At present, the interpretation gives wrong estimates of k . Also, data on DOP (p. 15) are left uninterpreted, likely due to the same reason. In particular, it is interesting to understand the reason why DOP at 681 and 90-nm-thick flakes are so low.

Response: We sincerely appreciate the reviewer's thoughtful reminder. We wholeheartedly agree with the reviewer's point regarding the importance of considering the reflection at the interfaces. Consequently, we have made the necessary revision to Equation 3, incorporating the transmittance amplitude in the following manner:

$$\vec{E} = E_0 \cdot \cos \theta \cdot t_x \cdot \cos(\omega t) \cdot \vec{x} + E_0 \cdot \sin \theta \cdot t_y \cdot \cos(\omega t - \delta) \cdot \vec{y} \quad (5)$$

where E_0 is the amplitude of incident light, θ is the angle between the incident light direction and the fast axis, t_x , t_y are the transmission coefficients, ω is the frequency, \vec{x} , \vec{y} are the unit vectors along the fast and slow axis.

Upon careful re-evaluation of the transmission light path, we found that the objective lens situated at the bottom of the confocal microscope was not perfectly aligned with the incident light path originating from the top. Consequently, we posit that the erroneous results of the Degree of Polarization (DOP) in our previous findings stemmed from the deviation in the optical path. We sincerely apologize for the inaccurate results provided earlier. To rectify this issue, we meticulously adjusted the light path and conducted comprehensive measurements on all the devices once again (Figs. R22-R26).

Fig. R22 **a** AFM images, and **b** the corresponding line profiles of the fibrous RP flakes with a thickness of 64 nm. **c** Polarization-dependent intensity of input/output spectra for the fibrous RP flake corresponding to Fig. R22a. **d** Schematic diagram of polarization state for input/output light corresponding to Fig. R22c.

Fig. R23 **a** AFM images, and **b** the corresponding line profiles of the fibrous RP flakes with a thickness of 100 nm. **c** Polarization-dependent intensity of input/output spectra for the fibrous RP flake corresponding to Fig. R23a. **d** Schematic diagram of polarization state for input/output light corresponding to Fig. R23c.

Fig. R24 **a** AFM images, and **b** the corresponding line profiles of the fibrous RP flakes with a thickness of 147 nm. **c** Polarization-dependent intensity of input/output spectra for the fibrous RP flake corresponding to Fig. R24a. **d** Schematic diagram of polarization state for input/output light corresponding to Fig. R24c.

Fig. R25 **a** AFM images, and **b** the corresponding line profiles of the fibrous RP flakes with a thickness of 178 nm. **c** Polarization-dependent intensity of input/output spectra for the fibrous RP flake corresponding to Fig. R25a. **d** Schematic diagram of polarization state for input/output light corresponding to Fig. R25c.

Fig. R26 **a** AFM images, and **b** the corresponding line profiles of the fibrous RP flakes with a thickness of 596 nm. **c** Polarization-dependent intensity of input/output spectra for the fibrous RP flake corresponding to Fig. R26a. **d** Schematic diagram of polarization state for input/output light corresponding to Fig. R26c.

In order to further explain the influencing factors of DOP value quantitatively, we added the following derivation in the revised supplementary information.

Fig. R27 Schematic diagram of the light vector decomposition

As depicted in Fig. R27, the incident light vector can be decomposed into components along the fast and slow axes of the fibrous RP. In order to streamline the model, we adopt the approximation that the b-axis of the fibrous RP represents the fast-axis. Consequently, upon traversing the fibrous RP flakes, the transmitted light vector can be expressed as follows:

$$\vec{E} = E_0 \cdot \cos \theta \cdot t_x \cdot \cos(\omega t) \cdot \vec{x} + E_0 \cdot \sin \theta \cdot t_y \cdot \cos(\omega t - \delta) \cdot \vec{y}. \quad (6)$$

The transmitted light vector along the fast and slow axes can be further decomposed along the direction of analyzer. The transmitter light vector after passing the analyzer can be expressed as follow:

$$\vec{E}_n = [E_0 \cdot \cos \theta \cdot t_x \cdot \cos \varphi \cdot \cos(\omega t) + E_0 \cdot \sin \theta \cdot t_y \cdot \sin \varphi \cdot \cos(\omega t - \delta)] \cdot \vec{n}, \quad (7)$$

where φ is the angle between the direction of analyzer and the fast axis. To simplify the formula, we define $A = E_0 \cdot \cos \theta \cdot t_x$ and $B = E_0 \cdot \sin \theta \cdot t_y$. Accordingly, by taking the time average within a cycle to eliminate t , we can obtain the expression for transmitted intensity I and angle φ :

$$\begin{aligned} I(\varphi) &= \frac{1}{T} \int_0^T |A \cdot \cos \varphi \cdot \cos(\omega t) + B \cdot \sin \varphi \cdot \cos(\omega t - \delta)|^2 dt \quad (\omega T = 2\pi) \\ &= \frac{1}{2} (A^2 \cdot \cos^2 \varphi + B^2 \cdot \sin^2 \varphi + 2AB \cdot \sin \varphi \cdot \cos \varphi \cdot \cos \delta). \end{aligned} \quad (8)$$

Polarization-dependent intensity of input/output spectra were well fitted by Equation 8 in the revised manuscript. By further simplification, we can get the following expression:

$$I(\varphi) = \frac{A^2+B^2}{4} + \frac{1}{2} \sqrt{\left(\frac{A^2-B^2}{2}\right)^2 + (AB\cos\delta)^2} \cdot \sin\left(2\varphi + \arctan\left(\frac{A^2-B^2}{2AB\cos\delta}\right)\right) \quad (9)$$

Thus, I_{max} and I_{min} can be obtained as follow:

$$I_{max} = \frac{A^2+B^2}{4} + \frac{1}{2} \sqrt{\left(\frac{A^2-B^2}{2}\right)^2 + (AB\cos\delta)^2}. \quad (10)$$

$$I_{min} = \frac{A^2+B^2}{4} - \frac{1}{2} \sqrt{\left(\frac{A^2-B^2}{2}\right)^2 + (AB\cos\delta)^2}. \quad (11)$$

Hence, we can get the expression of DOP:

$$DOP = \frac{I_{max}-I_{min}}{I_{max}+I_{min}} = \frac{2 \sqrt{\left(\frac{A^2-B^2}{2}\right)^2 + (AB\cos\delta)^2}}{A^2+B^2}. \quad (12)$$

Based on Equation 12, DOP is dependent on both the phase retarder value δ , and the angle θ between the incident light direction and the fast axis. Specifically, when the phase retarder value is fixed, DOP exhibits a correlation with θ . At $\theta = 0^\circ$ or 90° , DOP attains its maximum value of 1, indicating that the polarization state remains unchanged. Conversely, as θ approaches 45° , DOP reaches its minimum value.

To address the reviewer's concern, we have incorporated the transmittance amplitude in the Equation 5 in the revised manuscript, and calibrated the measurements of all the devices. Moreover, the above quantitative derivation of DOP has been added in the revised supplementary information, see Line 362-387 from Page 31 to Page 32.

In addition, Figs. R22, R26 have been added into the Fig. 5 in the revised manuscript. Figs. R23-R25, 27 have been added into the revised supplementary information as Supplementary Figs. 41-43, 40, respectively.

3. Does the spacing of Fabry-Perot peaks agree with other data on in-plane anisotropy like DFT calculations of n_1 , n_2 , and phase retardation estimations?

Response: We highly value the reviewer's considerate reminder and valuable suggestion. Initially, we attempted to extract the refractive index by analyzing the spacing of Fabry-Pérot peaks. However, after a simple calculation, we found that this approach does not yield accurate results for determining the refractive index. In Fig. R28, we present the PL spectrum of fibrous RP with a thickness of 1654

nm, illustrating the presence of the Fabry–Pérot effect.

Fig. R28 **a** PL spectrum of fibrous RP with Fabry–Pérot effect. **b** the corresponding line profiles of the fibrous RP flakes with a thickness of 1654 nm.

The Fabry–Pérot cavity related PL multi peaks are labeled as P1 (760nm), P2 (720 nm), P3 (689 nm), P4 (660nm), P5 (635nm), P6 (611 nm). According to $2dn_{peak} = k\lambda_{peak}$, we can obtain the following formulas:

$$\begin{aligned} 2 \cdot d \cdot n_{760 \text{ nm}} &= k \cdot \lambda_{760 \text{ nm}}, \\ 2 \cdot d \cdot n_{720 \text{ nm}} &= (k + 1) \cdot \lambda_{720 \text{ nm}}, \\ 2 \cdot d \cdot n_{689 \text{ nm}} &= (k + 2) \cdot \lambda_{689 \text{ nm}}, \\ 2 \cdot d \cdot n_{660 \text{ nm}} &= (k + 3) \cdot \lambda_{660 \text{ nm}}, \\ 2 \cdot d \cdot n_{635 \text{ nm}} &= (k + 4) \cdot \lambda_{635 \text{ nm}}, \\ 2 \cdot d \cdot n_{611 \text{ nm}} &= (k + 5) \cdot \lambda_{611 \text{ nm}}. \end{aligned}$$

If we set $k = 16$, then we can obtain the refractive index as follow:

$$\begin{aligned} n_{760 \text{ nm}} &= 3.676, \quad n_{720 \text{ nm}} = 3.700, \quad n_{689 \text{ nm}} = 3.749, \\ n_{660 \text{ nm}} &= 3.791, \quad n_{635 \text{ nm}} = 3.839, \quad n_{611 \text{ nm}} = 3.879. \end{aligned}$$

If we set $k = 17$, then we can obtain the refractive index as follow:

$$\begin{aligned} n_{760 \text{ nm}} &= 3.906, \quad n_{720 \text{ nm}} = 3.918, \quad n_{689 \text{ nm}} = 3.957, \\ n_{660 \text{ nm}} &= 3.990, \quad n_{635 \text{ nm}} = 4.031, \quad n_{611 \text{ nm}} = 4.063. \end{aligned}$$

If we set $k = 18$, then we can obtain the refractive index as follow,

$$\begin{aligned} n_{760 \text{ nm}} &= 4.135, \quad n_{720 \text{ nm}} = 4.135, \quad n_{689 \text{ nm}} = 4.166, \\ n_{660 \text{ nm}} &= 4.190, \quad n_{635 \text{ nm}} = 4.223, \quad n_{611 \text{ nm}} = 4.248. \end{aligned}$$

In situations where the determination of the positive integer k value becomes uncertain, multiple sets of potential refractive index values arise, rendering this method impractical. As recommended by Reviewer 2, we performed a fresh measurement of the polarized PL spectrum after reducing the sample thickness to below 300 nm, effectively eliminating the occurrence of the Fabry–Pérot effect. To avoid any potential ambiguity, we have removed the mention of the Fabry–Pérot phenomenon and its correlation with the refractive index in the revised manuscript.

Furthermore, it is important to note that the phase retardation value δ solely provides the in-plane refractive index difference (Δn) at 532 nm and does not yield the precise values for refractive index (n_1, n_2). As explained in comment 2, the phase retardation value δ can be derived using Equation 13:

$$\cos^2\delta = \frac{DOP^2(A^2+B^2)^2 - (A^2-B^2)^2}{4A^2B^2}. \quad (13)$$

Combining $\delta = \Delta n 2\pi d / \lambda$, we can further obtain the in-plane refractive index difference at 532 nm. The extracted constants are exhibited in the following Table R1:

Table R1 The correlation constants obtained by extraction.

d (nm)	DOP	δ	Δn	$\Delta \bar{n}$	Δn_{cal}
64	0.995	0.185792	0.245798		
100	0.981	0.28811	0.243944		
147	0.916	0.488021	0.281095	0.26171 ± 0.02359	0.513
178	0.873	0.514115	0.244552		
596	0.8	2.063419	0.293139		

Consequently, through phase retardation estimation, we have calculated $\Delta n = 0.26171 \pm 0.02359$. It is important to note that this value is merely half of the theoretical value. There are a couple of factors contributing to this discrepancy. Firstly, theoretical calculations tend to overestimate the refractive index. Secondly, during the testing process, various factors such as the hypothesis incorporated in the formula derivation and systematic errors in signal acquisition can introduce uncertainties that impact the accuracy of the Δn obtained through phase retardation estimation. These considerations highlight the complexities involved in obtaining precise refractive index values.

To clarify, we have added “Additionally, we determined the corresponding phase retarder value δ and the in-plane refractive index difference $\Delta n'$. The summarized results can be found in Supplementary Table 5. Consequently, through phase retardation estimation, we calculated $\Delta n'$ to be 0.26171 ± 0.02359 . It's important to note that this value is only half of the theoretical value. Several factors contribute to this discrepancy. Firstly, theoretical calculations tend to overestimate the refractive index. Secondly, uncertainties arising from factors like the hypothesis incorporated in the formula derivation and systematic errors in signal acquisition during the testing process can impact the accuracy of $\Delta n'$ obtained through phase retardation estimation. These factors highlight the complexities involved in obtaining precise refractive index values.”, see Line 706-717 on Page 31 in the revised manuscript with revision traces.

In addition, the Table R1 has been added in the revised supplementary information as Supplementary Table 5.

4. It is mentioned that the reported in-plane anisotropy is greater than that of As₂S₃, but it seems to be high in the high absorption region, which the authors do not explicitly state. There is also another nuance: for some reason, authors do not provide experimental data in the spectral range but show only DFT results. The authors do not seem to extract any constants from the polarization transmission spectra. Instead, they pointed out that extracting the refractive index for specific wavelengths is possible using photoluminescence spectra and even wrote a formula. Still, I can not see any numbers

or explicit indications of experimental data (only DFT is given).

Response: We express our gratitude to the reviewer for providing a valuable suggestion. When light interacts with solids, the electric field component of the electromagnetic wave induces anisotropic dielectric polarization in materials with low symmetry. This anisotropy of electric polarization, in turn, gives rise to optical anisotropy through the manifestation of the anisotropic dielectric function [Ref. 21: *Nanoscale* **14**, 12238 (2022) and Ref. 22: *Adv. Mater.* **29**, 1700441 (2017)]. In the high absorption region, photon absorption, electron transitions, and other processes make the interaction between light and solids more intricate, resulting in a greater degree of anisotropy in the refractive index within that particular region. In traditional birefringent bulk optical crystals, there is also a sharp decline followed by a steady trend in the birefringence values before and after the high absorption region [*Angew. Chem. Int. Ed.* **58**, 17675-17678 (2019), *Angew. Chem. Int. Ed.* **60**, 3464-3468 (2021), *Angew. Chem. Int. Ed.* **60**, 3530-3544 (2021), *Angew. Chem. Int. Ed.* **61**, e202202746 (2022) and *Angew. Chem. Int. Ed.* **134**, e202208811 (2022)]. However, in order to minimize the losses caused by optical absorption, it is common to report and utilize the birefringence values within the low absorption region. It should be noted that for low-dimensional materials, due to thickness constraints, even in the high absorption region, the optical losses are still limited. Based on these reasons, we provided the birefringence values in that high absorption region in the original manuscript.

Furthermore, the extraction of constants through Fabry–Pérot effect or phase retardation estimation spectra have been discussed in response to Comment 2. We will subsequently explain why the experimental determination of the refractive index is not included in this article. For conventional optical bulk crystals, the measurement of their refractive index is typically accomplished using ellipsometer technology. However, there are currently no commercially available testing equipments capable of experimentally determining the refractive index of materials at the micrometer scale. In previous literature, there are primarily two methods for determining the refractive index of low-dimensional materials. The first approach is Kramers–Kronig (KK) analysis of the reflection or transmission spectrum of the sample [*Phys. Rev. B* **90**, 205422 (2014)]. The second approach involves measuring the contrast values of materials at different thicknesses through micro-area contrast spectroscopy and obtaining the refractive index through numerical fitting [*J. Am. Chem. Soc.* **138**, 300-305 (2016), *Nanoscale* **10**, 12424-12429 (2018) and *Adv. Optical Mater.* **7**, 1900239 (2019)].

In our study, we have indeed conducted experimental measurements of the refractive index of fibrous RP using micro-area contrast spectroscopy and micro-area ellipsometer imaging technology. The obtained birefringence values surpass those reported recently for experimental measurements of Te. We intend to report on this aspect in future work. Additionally, considering that the focus of this work lies in the first success in substrate-growth of fibrous RP flakes, growth mechanism, and the first device demonstration of the fibrous RP flakes, delving extensively into refractive index testing and discussion would shift the focus away from the main theme of the article. Therefore, in this particular work, we have chosen to solely present the refractive index of fibrous RP calculated through DFT.

To address the reviewer’s concern, we have added “*In the high absorption region, two peaks of birefringence ($\Delta n = n_z - n_x$) are observed, with values of 1.646 @ 416 nm and 1.382 @ 592 nm, respectively. Subsequently, there is a sharp decline followed by a consistent trend in the birefringence values, which remains at approximately 0.5 in the near-infrared region. This behavior can be attributed to the intricate interaction between light and solids in the high absorption region, where processes such as photon absorption and electron transitions contribute to a greater degree of anisotropy in the refractive index. Furthermore, the xoy-plane refractive index differences ($\Delta n' = n_y - n_x$) exhibit remarkable characteristics, with a value of 0.739 @ 467 nm and consistently large values (above 0.3) across a wide spectral range (see Supplementary Fig. 36b).*”, in the revised

manuscript with revision traces, see Line 546-561 on Page 24.

5. *Side remark: I noticed multiple uses of "angle-resolution polarized photoluminescence", shouldn't it be "angle-resolved"?*

Response: We really appreciate the reviewer's kind reminder. We have corrected the expression and changed all the "*angle-resolution polarized*" to "*angle-resolved polarized*" in the revised manuscript.

Once again, we would like to convey our appreciation to the Reviewer 3 for his time and effort invested in reviewing our manuscript and providing thoughtful comments that contributed to its refinement.

REVIEWERS' COMMENTS

Reviewer #1 (Remarks to the Author):

The author has revised the article according to the requirements and made great improvements. This manuscript can be accepted.

Reviewer #2 (Remarks to the Author):

The authors have addressed this reviewer's questions. Noted that more deep discussion and essential experiments have been added. It could be accepted.

Reviewer #3 (Remarks to the Author):

I am satisfied with the authors' reply.
The authors addressed all my comments and recommendations.